# Integrin α3β1 in hair bulge stem cells modulates CCN2 expression and promotes skin tumorigenesis

Veronika Ramovs[1] , Ana Krotenberg Garcia[1], Ji-Ying Song[2], Iris de Rink[3], Maaike Kreft[1], Roel Goldschmeding[4] , Arnoud Sonnenberg[1]

Epidermal-specific deletion of integrin α3β1 almost completely prevents the formation of papillomas during 7,12-Dimethylbenz [a]anthracene/12-O-tetradecanoylphorbol-13-acetate (DMBA/TPA) two-stage skin carcinogenesis. This dramatic decrease in tumorigenesis was thought to be due to an egress and premature differentiation of α3β1-depleted hair bulge (HB) stem cells (SCs), previously considered to be the cancer cells-of-origin in the DMBA/TPA model. Using a reporter mouse line with inducible deletion of α3β1 in HBs, we show that HB SCs remain confined to their niche regardless of the presence of α3β1 and are largely absent from skin tumors. However, tumor formation was significantly decreased in mice deficient for α3β1 in HB SCs. RNA sequencing of HB SCs isolated from short-term DMBA/TPA–treated skin showed α3β1-dependent expression of the matricellular protein connective tissue growth factor (CCN2), which was confirmed in vitro, where CCN2 promoted colony formation and 3D growth of transformed keratinocytes. Together, these findings show that HBs contribute to skin tumorigenesis in an α3β1-dependent manner and suggest a role of HB SCs in creating a permissive environment for tumor growth through the modulation of CCN2 secretion.

## Introduction

Hair bulge (HB) contains one of the most studied and characterized stem cell (SC) compartments in mammalian skin. Located at the bottom of the hair follicles (HFs) in the resting stage of hair cycle (i.e., telogen), it is the main reservoir for cells that form elongated HFs during the growth phase of hair cycle called anagen (Cotsarelis, 2006; Purba et al, 2014; Rompolas & Greco, 2014). The transient-amplifying progeny of HB SCs normally remains confined within HFs; however, under specific circumstances, some of these cells can egress and contribute to the interfollicular epidermis (IFE). Such behavior has been observed during wound healing, when HB

keratinocytes migrate out of their niche into the newly formed epidermis (Ito et al, 2005; Vagnozzi et al, 2015).

It has been suggested that HB SCs also play a crucial role in cutaneous skin tumorigenesis. An increased expression of HB marker keratin 19 (K19) has been observed in human squamous cell carcinomas (SCCs) (Chen et al, 2008), whereas basal cell carcinomas and trichoblastomas were shown to up-regulate the expression of the HB marker keratin 15 (K15) (Kim et al, 2016). Furthermore, several studies have suggested that HB SCs represent the cells-of-origin of papillomas and benign tumors that can progress into invasive SCCs, and are formed during the two-stage chemically induced mouse skin carcinogenesis protocol (7,12-Dimethylbenz[a]anthracene/12-O-tetradecanoylphorbol-13-acetate [DMBA/TPA] treatment) (Kim et al, 2009; Lapouge et al, 2011; Li et al, 2013; Sánchez-Danés & Blanpain, 2018). The suggestion that HB SCs play a crucial role in DMBA/TPA–induced carcinogenesis was also made in our previously reported study in mice, lacking integrin α3β1 in epidermis (K14 Itga3 KO mice). Integrin α3β1 is a transmembrane receptor for laminins-332 and laminins-511 in the epidermal basement membrane and functions as bidirectional signaling molecule (Subbaram & Dipersio, 2011; Ramovs et al, 2017). Upon its epidermal deletion, mice exhibit an increased epidermal turnover, which coincides with the loss of label-retaining cells and, importantly, with the localization of K15-positive keratinocytes in IFE and upper parts of HFs (i.e., isthmus and infundibulum) (Sachs et al, 2012). As K14 Itga3 KO mice showed a near absence of DMBA/TPA–induced tumorigenesis, we hypothesized that this could be due to the egress of DMBA-primed K15-positive HB SCs and their loss through squamous differentiation in IFE (Sachs et al, 2012). However, the HB origin of DMBA/TPA–derived tumors is somewhat controversial. Recent studies reported a limited contribution of HB SCs to papillomas (van de Glind et al, 2016) and demonstrated the importance of SCs residing in isthmus instead (Huang et al, 2017). Furthermore, it has been reported that K15 promoter, which is widely used to generate genetic deletions in HB SCs, also targets basal cells in IFE, isthmus, and infundibulum (Lapouge et al, 2011) and, importantly, that K15

[1]Division of Cell Biology, The Netherlands Cancer Institute, Amsterdam, The Netherlands [2]Department of Experimental Animal Pathology, The Netherlands Cancer Institute, Amsterdam, The Netherlands [3]Genomics Core Facility, The Netherlands Cancer Institute, Amsterdam, The Netherlands [4]Department of Pathology, University Medical Center Utrecht, Utrecht, The Netherlands

Correspondence: a.sonnenberg@nki.nl
Ana Krotenberg Garcia's present address is Division of Molecular Pathology, The Netherlands Cancer Institute, Amsterdam, The Netherlands

can be mis-expressed during tumorigenesis, reflecting the activity and responsiveness of basal epidermal cells to the loss of skin homeostasis (Troy et al, 2011). Considering all this, we found it important to re-evaluate the mechanisms behind the absence of tumorigenesis in K14 *Itga3* KO mice. Here, we exploit linage tracing and next-generation sequencing analysis of a mouse model with inducible deletion of α3β1 in HBs to investigate the role of α3β1 in HB SCs and their contribution to skin tumorigenesis.

## Results

### HB keratinocytes lacking integrin α3β1 stay confined within their niche and contribute normally to hair cycle

To target and visualize HB SCs and their progeny, we generated a mouse line, expressing an inducible Cre (CreER) under the HB-specific K19 promoter (K19-CreER) (Means et al, 2008) in combination with mT/mG reporter transgene (Muzumdar et al, 2007) (K19 *Itga3* WT mice). To investigate the role of α3β1 in HB SCs, we have further introduced floxed *Itga3* alleles to our mouse model (K19 *Itga3* KO mice) (Fig 1A), which resulted in an efficient deletion of α3β1 in Cre-induced GFP-positive cells (Fig 1B). Tamoxifen administration induced GFP expression in most HFs in both K19 *Itga3* KO and WT mice (Fig S1A) and, as expected, the GFP-positive cells localized to HBs, with no obvious leakiness detected (Fig 1C–E). Surprisingly, linage tracing of HB cells, induced in the first telogen phase of the hair cycle (P21), showed that HB SCs remain confined to their niche, regardless of the presence of α3β1, and do not egress into infundibulum and/or IFE (Figs 1C and E and S1B), as was expected based on previously reported observations of K15-positive cells in the IFE of mice with an epidermis-specific deletion of α3β1 (K14 *Itga3* KO mice) (Sachs et al, 2012). Furthermore, linage tracing during the first hair cycle revealed that *Itga3* KO and WT HB cells contribute to all layers of growing HFs, with no observed differences in hair cycle progression between K19 *Itga3* KO and WT mice (Fig 1E). Taken together, these data demonstrate that α3β1 plays no major role in HBs of adult mice under homeostatic conditions.

### Epidermal deletion of α3β1 causes de novo expression of K15 outside of HBs

The finding that HB cells do not egress from their niche into the IFE of K19 *Itga3* KO mice casted doubt on whether the K15-positive cells, previously observed in the IFE of K14 *Itga3* KO mice, truly originate from HBs (Sachs et al, 2012). To re-evaluate their origin, we turned to the K14 *Itga3* KO and WT mouse models (Fig 2A). In accordance with previous work on this mouse model, immunostaining of whole-mount tail epidermis from K14 *Itga3* KO mice confirmed the presence of K15-positive and α3β1-depleted keratinocytes in isthmus, infundibulum, and IFE (Sachs et al, 2012) (Fig 2B). Remarkably, we observed that α3β1-positive cells that had escaped Cre-recombinase in K14 *Itga3* KO mice preferentially localized to HBs in both tail and back skin (Fig 2B and C). We confirmed this non-stochastic localization with flow cytometry, which showed that the remaining α3-positive keratinocytes isolated from back epidermis of K14 *Itga3* KO mice were twice more likely to originate from HBs

than their counterpart isolated from WT mice (Figs 2D and S2A). The relatively inefficient deletion of α3β1 in HBs in K14 *Itga3* KO mice together with the absence of mis-localized HB keratinocytes in K19 *Itga3* KO mice suggests that the loss of α3β1 in HB SCs does not lead to their egress, but rather that the mis-localized K15-positive keratinocytes originate from other epidermal compartments of K14 *Itga3* KO mice. To determine whether K15 in these cells could be expressed de novo upon α3β1 deletion, we performed RT-qPCR of the RNA, isolated from tail and back epidermis of K14 *Itga3* KO and WT mice. Indeed, an increased expression of K15 could be detected in the epidermis of K14 *Itga3* KO, compared with WT mice (Fig 2E), despite the absence of proliferating K15-positve cells residing in HBs (Fig S2B). We further confirmed that the mis-localized K15-positive keratinocytes were not derived from HB SCs by performing a quantitative flow cytometry analysis of the HB population size (CD34+, α6high), which was comparable between K14 *Itga3* KO and WT mice (Fig S2C and D). This remained true even after mice were submitted to the short-term DMBA/TPA treatment, mimicking the initiation stage of tumorigenesis, during which an increased miss-localization of K15-positive keratinocytes had been previously reported for K14 *Itga3* KO mice (Sachs et al, 2012) (Fig S2C and D). All in all, these data strongly suggest that keratinocytes in IFE and upper parts of HFs express K15 de novo in the absence of α3β1 and provide new evidence that HB SCs remain confined within their niche regardless of whether they express α3β1.

### The contribution of HB keratinocytes to newly formed IFE is increased in the absence of α3β1

The egress of HB SCs into IFE normally occurs during wound healing, when HB keratinocytes contribute to the formation of neo-epidermis (Ito et al, 2005; Vagnozzi et al, 2015). To test whether our model reflects these characteristics of HB SCs and to determine whether the presence of α3β1 affects induced egress of HB keratinocytes, we performed wounding experiments of K19 *Itga3* KO and WT mice. The presence of α3β1 in HBs did not affect the closure of the wounds; no differences were observed in the length of neo-epidermis 3 d after the wounding and in the percentage of closed wounds 5 d after wounds were inflicted (Fig 3A and B). As expected, Cre-induced, GFP-positive HB keratinocytes contributed to the formation of neo-epidermis upon wounding of both K19 *Itga3* KO and WT mice (Fig 3C and D). Whereas at the beginning of the wound re-epithelization, no obvious differences in the contribution of HB-originating cells to the neo-epidermis could be observed between K19 *Itga3* KO and WT mice (Fig 3C), there was a small, but significant increase in the number of GFP-positive cells in the neo-epidermis of K19 *Itga3* KO mice at the final stages of wound closure (Fig 3D). This is consistent with previous observations that the absence of α3β1 promotes cell migration during wound healing (Margadant et al, 2009). Together, these findings indicate that during wound healing, the egress of HB SCs into the neo-epidermis increases when α3β1 is absent from HB keratinocytes, which, however, does not alter the rate of re-epithelization.

### The absence of α3β1 in HBs reduces susceptibility of mice to DMBA/TPA–mediated tumorigenesis

Next, we investigated the role of HB SC–residing α3β1 in skin carcinogenesis by submitting K19 *Itga3* KO and WT mice to the

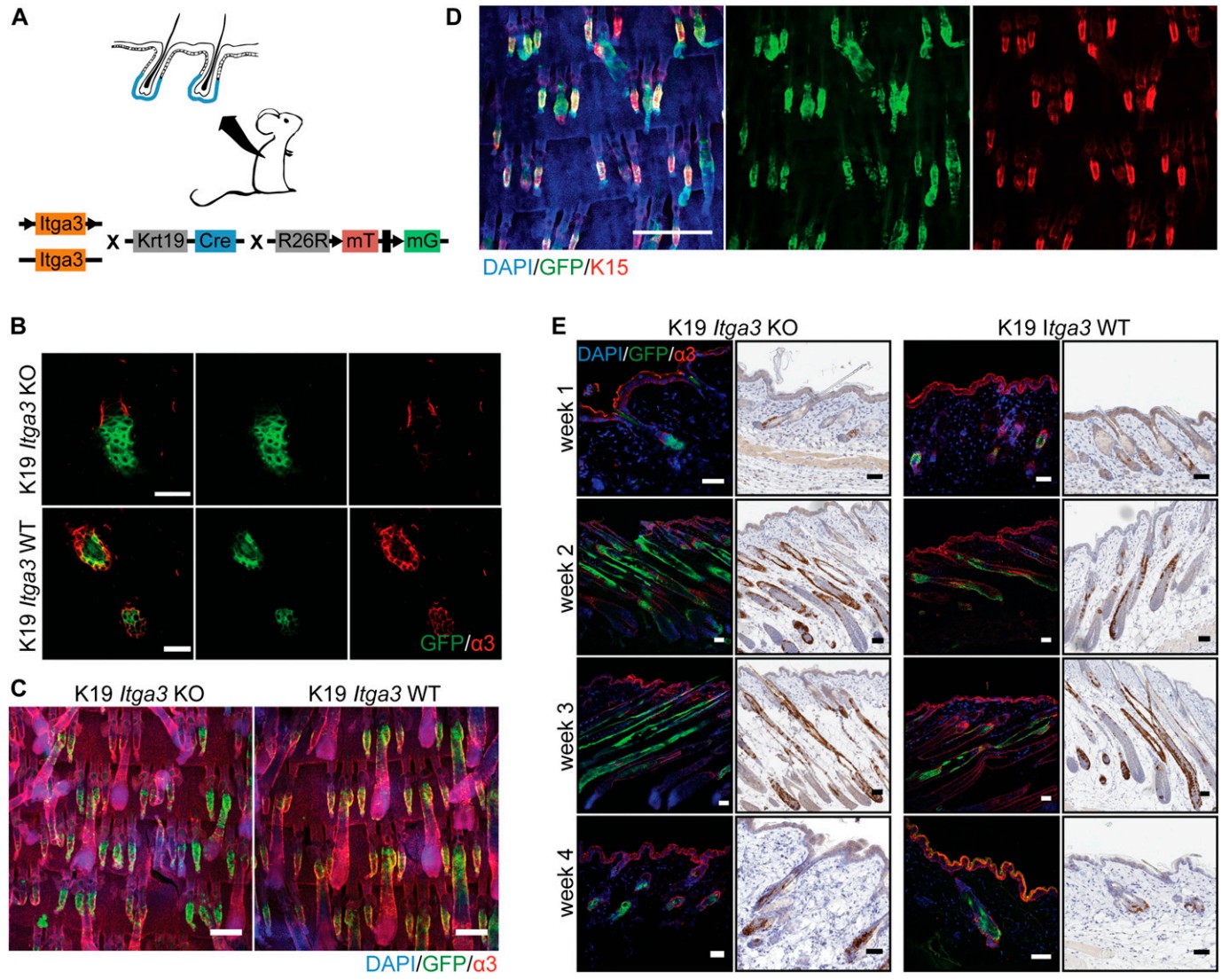

**Figure 1.  HB keratinocytes lacking integrin α3β1 stay confined within their niche and contribute normally to hair cycle.**
**(A)** Overview of the K19 *Itga3* KO and WT mouse models. **(B)** Integrin α3β1 is expressed in Cre-induced GFP-positive keratinocytes of K19 *Itga3* WT mice and efficiently deleted in GFP-positive keratinocytes of 7-wk-old K19 *Itga3* KO mice 1 wk after tamoxifen treatment (back skin, scale bar: 30 μm). **(C)** Linage tracing of GFP-positive HB keratinocytes showing localization within their niche in K19 *Itga3* KO and WT mice 1 wk after tamoxifen treatment (whole mounts of tail epidermis, scale bar: 200 μm). Linage tracing of up to 4 wk can be found in Fig S1B. **(D)** Whole mount of tail epidermis of K19 *Itga3* WT mouse showing co-localization of Cre-induced GFP-positive cells and K15 marker in HBs (scale bar: 500 μm). **(E)** Linage tracing of GFP-positive HB SCs Cre-induced in telogen (P21) and followed for up to 4 wk over whole hair cycle (until P49) in the back skin of K19 *Itga3* KO and WT mice. Representative images of two to three mice per condition are shown (scale bar: 50 μm).

complete DMBA/TPA carcinogenesis protocol. Tumors could be detected 6 wk after the beginning of the treatment (P91-P97), regardless of the presence of α3β1 (Fig 4A). Although both K19 *Itga3* KO and WT mice developed numerous tumors by the end of the treatment (K19 *Itga3* KO 33.8 and WT 49.1 tumors on average), there was a marked 30% decrease in tumor formation and a 15% decrease in the average tumor size upon the deletion of *Itga3* in HB SCs (Figs 4A and B and S3A). In line with this, K19 *Itga3* KO mice exhibited significantly lower tumor burden than WT mice (K19 *Itga3* KO 1313.5 and WT 2296.1 mm$^2$ on average, Fig 4A). Nearly all the tumors were benign papillomas and keratoacanthomas, with no notable difference in their prevalence between the two mouse lines (Fig 4C). By the end of the treatment (P196), ulcerating tumors were observed in

one K19 *Itga3* KO and three WT mice, which were identified as SCCs (K19 *Itga3* KO and WT) and keratoacanthomas with carcinomatous changes (K19 *Itga3* WT) by histological analysis (Fig S3B). As this incidence was too low to draw any conclusions about the role of α3β1 in the malignant progression of tumors, we selected seven K19 *Itga3* KO and seven WT mice with low tumor burden and treated them with TPA for up to an additional 10 wk, until they had to be euthanized because of the tumor burden or ulceration of tumors. Tumor progression was more commonly observed in K19 *Itga3* KO (five of seven) than in K19 *Itga3* WT mice (three of seven). Furthermore, K19 *Itga3* KO mice developed also high malignancy–grade tumors such as spindle cell sarcoma and mixed basal SCC in addition to the SCCs and keratoacanthomas with carcinomatous

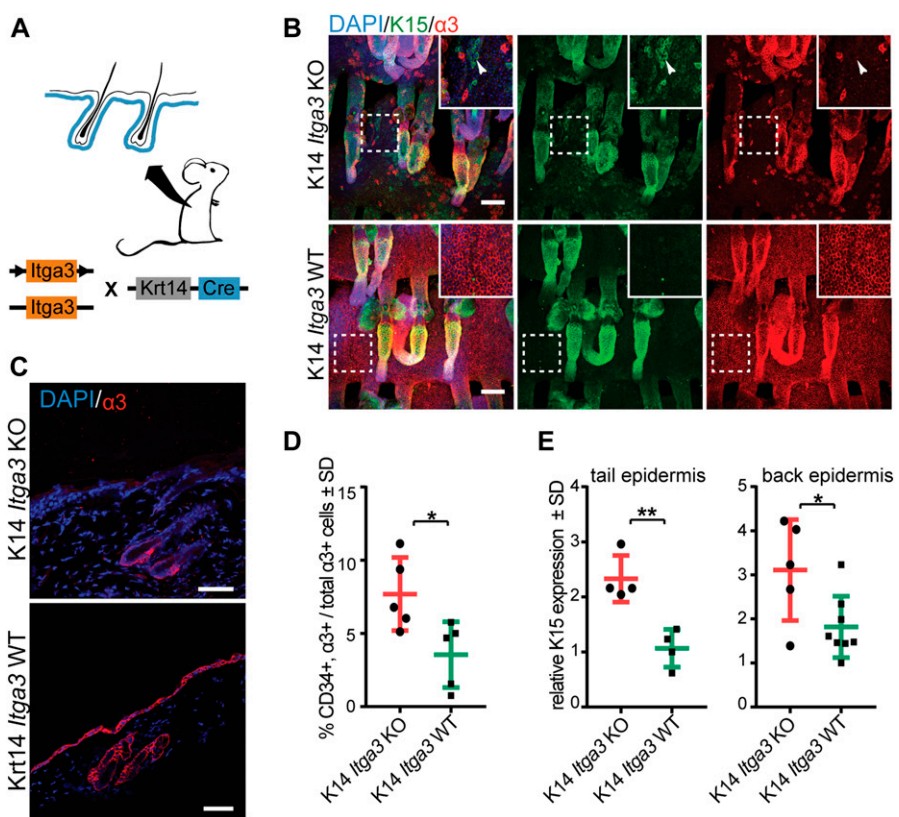

**Figure 2. Epidermal deletion of α3β1 causes de novo expression of K15 outside of HBs.**
**(A)** Overview of the K14 *Itga3* KO and WT mouse models. **(B)** Whole mounts of tail epidermis show the presence of α3β1-depleted K15-positive keratinocytes in upper parts of hair follicles and IFE of K14 *Itga3* KO mice (white arrow heads). Remaining α3β1-positive keratinocytes in K14 *Itga3* KO mice are preferentially localized to HBs (scale bar: 100 μm). **(C)** Staining for integrin α3 shows HB localization of α3β1-positive keratinocytes in the back skin of 7-wk-old K14 *Itga3* KO mice. α3β1 is found in all basal keratinocytes of K14 *Itga3* WT mice of similar age (scale bar: 50 μm). **(D)** FACS analysis of keratinocytes isolated from back skin epidermis. The chart shows the percentages of α3-positive HB cells (CD34-positive) in the total α3-positive population. Each dot represents a mouse. Gating strategy can be found in Fig S2A (mean ± SD, unpaired *t* test, *P < 0.05). **(E)** GAPDH-normalized relative mRNA expression of K15 is increased in the epidermis of back and tail skin of K14 *Itga3* KO compared with WT mice. Each dot represents a mouse and is an average of technical duplicate or triplicate (mean ± SD, unpaired *t* test, *P < 0.05, **P < 0.005). Source data are available for this figure.

changes (Fig S3C). Although the yield of progressed tumors after this prolonged treatment was still too low to draw firm conclusions, the observed trend fits with the results of tumorigenesis experiments previously performed with K14 *Itga3* KO and WT mice (Sachs et al, 2012). All in all, these data show that α3β1 in HB SCs promotes formation of benign tumors during DMBA/TPA treatment.

### HB-derived keratinocytes are largely absent from skin tumors

The moderate reduction of tumorigenesis in the K19 *Itga3* KO mice, compared with the near complete absence of tumor formation in mice carrying a targeted deletion of *Itga3* in the whole epidermis, suggests that HB SCs might be the cells-of-origin for some, but not all DMBA/TPA–initiated tumors. To determine whether papillomas can arise from Cre-initiated HB keratinocytes in K19 *Itga3* KO and WT mice, we have analyzed the cross-sectional areas of 321 and 365 tumors, respectively, for the presence of GFP. Remarkably, most tumors were negative for GFP, and only one small papilloma isolated from K19 *Itga3* WT mouse consisted almost entirely of GFP-positive cells (Fig 5A and B). Consistent with their non-HB origin, all the analyzed tumors stained positive for α3β1 (Fig 5C). 2.8% of tumors, isolated from the K19 *Itga3* KO mice and 8.2% of the K19 *Itga3* WT tumors contained patches of GFP-positive cells, accounting to up to 5% of the total tumor area (Fig 5B). Such minor cell populations in otherwise monoclonal DMBA/TPA–derived papillomas have been recently described and shown to lack the activating mutation in *Hras*, the predominant proto-oncogene

activated in DMBA/TPA–induced tumors (Reeves et al, 2018). As the assessment of the amount of HB-originating GFP-positive cells in the tumors was determined based on only one cross section, we reasoned that the actual numbers are likely higher. Indeed, an analysis of 10 cross sections (200-μm step size) in randomly selected tumors of four K19 *Itga3* KO and four WT mice showed that, respectively, 9.6% and 32.3% of the tumors contain regions of GFP-positive cells. These GFP-positive regions occupy less than 0.1% of the tumor area in the majority of tumors (Fig 5D and E). The number of tumors, containing GFP-positive cells and the GFP-positive regions were significantly reduced in tumors, originating from K19 *Itga3* KO, compared with WT mice (Fig 5D). Taken together, these data demonstrate that HB SCs are not the main tumor-initiating cells in the two-stage carcinogenesis model and that HB-derived keratinocytes constitute a minor cell population in some tumors.

### α3β1-depleted keratinocytes show an increased differentiation signature and decreased expression of CCN2 during the initiation stage of tumorigenesis

The non-HB origin of most tumors together with the decreased tumorigenesis upon the deletion of HB-derived *Itga3* indicate that α3β1 in HB SCs likely promotes tumorigenesis indirectly, that is, through changes in the pro-tumorigenic environment or by directly affecting neighboring keratinocytes. Because the deletion of *Itga3* in HB SCs had a larger effect on the initiation of tumorigenesis than on the rate of tumor growth (Figs 4A and S3A), we investigated a

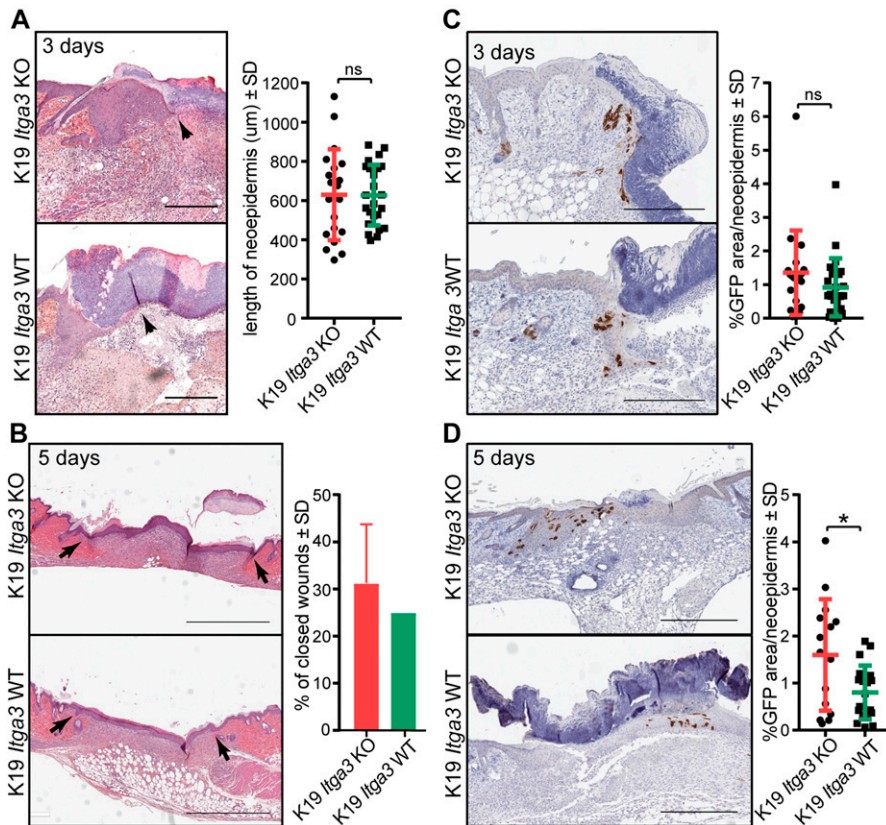

**Figure 3.  The contribution of HB keratinocytes to newly formed IFE is increased in the absence of α3β1.**

**(A, B)** H&E staining (left) and quantification (right) of wound healing. **(A, B)** Wound closure is comparable between K19 *Itga3* KO and WT mice, 3 (A) and 5 d (B) after wounding. **(A)** Each dot represents the average length of the neo-epidermis (black arrows) per wound (mean ± SD, unpaired *t* test). Wounds of six K19 *Itga3* WT and five K19 *Itga3* KO mice were analyzed (scale bar: 300 *μ*m). **(B)** Bars represent the percentage of closed wounds per mouse (mean ± SD). Wounds of four K19 *Itga3* WT and four K19 *Itga3* KO mice were analyzed (scale bar: 1 mm). **(C, D)** Immunohistochemistry (IHC) staining for GFP (left) and quantification (right) of GFP-positive area per neo-epidermis of K19 *Itga3* KO and WT mice. **(C)** HB-originating GFP-positive keratinocytes comparably contribute to neo-epidermis in K19 *Itga3* KO and WT mice 3 d after wounding (scale bar: 300 *μ*m). Each dot represents the percentage of GFP-positive area per wound (mean ± SD, unpaired *t* test). Wounds of six K19 *Itga3* WT and five K19 *Itga3* KO mice were analyzed. **(D)** 5 d after the wounding, the contribution of the α3β1-deficient HB SCs to the newly formed epidermis is more extensive than that of the α3β1-proficient HB SCs. Each dot represents the percentage of GFP-positive area per wound (mean ± SD, unpaired *t* test, *P < 0.05). Wounds of five K19 *Itga3* WT and four K19 *Itga3* KO mice were analyzed. Source data are available for this figure.

potential role of α3β1 in establishing a tumor-supportive environment. To this end, we performed RNA sequencing of GFP-positive keratinocytes isolated from the skin of K19 *Itga3* KO and WT mice during the initiation stage of tumorigenesis induced by short-term DMBA/TPA treatment, which has been shown to be sufficient for the outgrowth of papillomas (Diwan et al, 1985; Hennings et al, 1985). This stage of the two-stage chemical carcinogenesis, when pro-tumorigenic pathways are switched on, but tumors have not yet been formed, thus reflects the cell environment that can support tumor formation. At this time point, HB-originating GFP-positive cells can be found in both outer and inner layers of the growing HFs and, interestingly, in some cases in the isthmus, infundibulum, and IFE of K19 *Itga3* KO and WT mice (Fig 6A). Gene expression profiling of the GFP-positive keratinocytes by RNA sequencing confirmed their HB origin (e.g., high expression of CD34, Lgr5, and K15) (Fig S4A). A total of 15 protein-coding genes were significantly differentially expressed between the GFP-positive *Itga3* KO and WT keratinocytes. Several of the hits that displayed an increased expression in *Itga3* KO keratinocytes are known to be involved in squamous cell differentiation (Fig 6B), which is in line with previously reported observations of the increased epidermal turnover upon the deletion of *Itga3* (Sachs et al, 2012). Importantly, several of the hits belonged to the keratinocyte secretome, indicating that α3β1 could promote the formation of tumor-permissive environment through regulation of paracrine signaling of the HB SCs (Fig 6B). The high expression of connective tissue growth factor

CCN2 (also CTGF) was of particular interest, as this protein has a broad regulatory function in a variety of important biological and pathological processes, is a known integrin interactor, and has been implemented in skin tumorigenesis before (Quan et al, 2014; Ramazani et al, 2018). RNA sequencing data were validated using immunohistochemical and immunofluorescent (IF) staining, which confirmed that CCN2 localizes to HFs and HB SCs of short-term DMBA/TPA–treated back skin (Figs 6C and S4B), as has been previously reported (Rittié et al, 2009; Liu & Leask, 2013). Furthermore, IF staining confirmed the reduction of CCN2 expression in GFP-positive HFs lacking α3β1 (Fig 6C). In papillomas, CCN2 expression could be observed in isolated epithelial or stromal cells and occasionally in cell clusters, which did not correlate to the HB-originating GFP-positive areas in consecutive sections of tumors, isolated from K19 *Itga3* WT mice (Fig S4C and D). Although CCN2 could be detected in all K19 *Itga3* WT tumors analyzed, a small number of CCN2-positive cells (the mean CCN2-positive surface is 0.27% of total tumor area) together with their non-HB origin indicates that CCN2 does not play a major role in the late stages of tumor growth (Fig S4C).

Together, these data show that during the initiation stage of tumorigenesis, α3β1 in HB SCs suppresses HF differentiation and regulates the expression of CCN2 and several other proteins that are part of HB SC secretome. Thus, α3β1 might affect the cell environment during early tumorigenesis through regulation of the paracrine signaling.

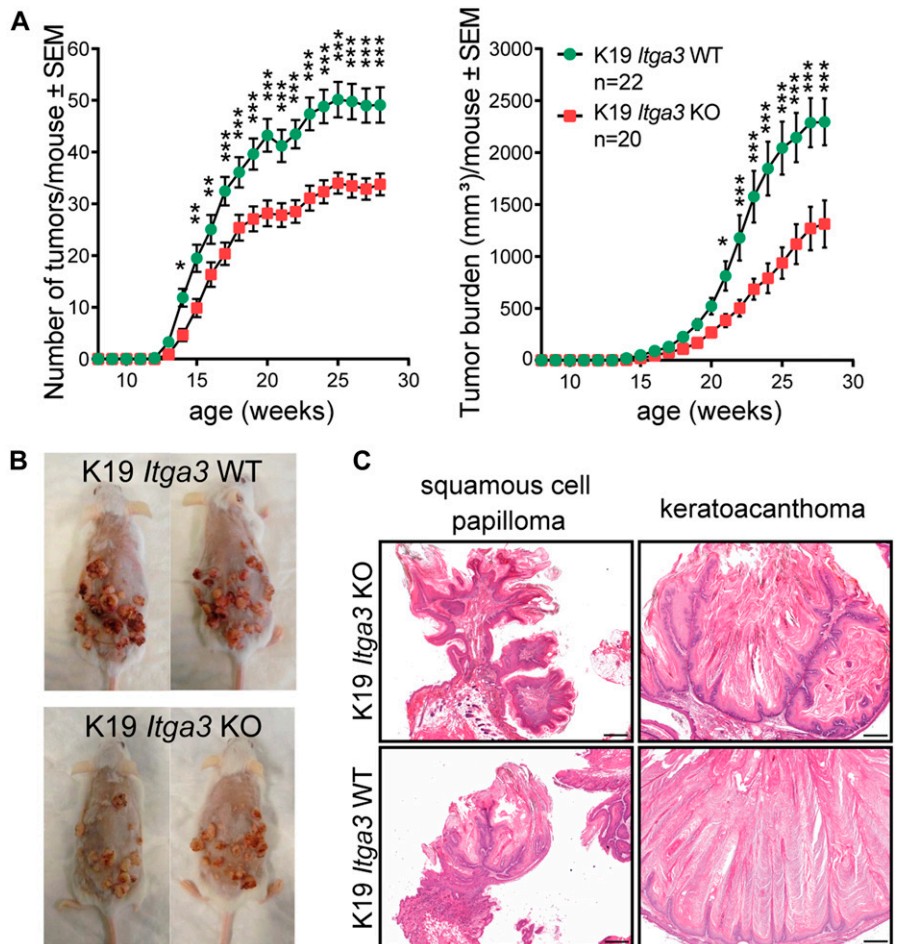

**Figure 4.   The absence of α3β1 in HBs reduces susceptibility of mice to DMBA/TPA–mediated tumorigenesis.**
**(A)** The number of tumors (left) and tumor burden (right) is decreased in K19 *Itga3* KO compared with WT mice submitted to the DMBA/TPA carcinogenesis protocol (mean ± SEM, unpaired *t* test, *P < 0.05, **P < 0.005, ***P < 0.0005). **(B)** Representative macro images of K19 *Itga3* KO and WT mice at the end of the treatment. **(C)** Histology of benign papillomas and keratoacanthomas, representing the majority of tumors isolated from K19 *Itga3* KO and WT mice. Source data are available for this figure.

## CCN2 expression is α3β1 dependent and promotes colony formation and 3D growth of *Hras*-transformed keratinocytes in vitro

Next, we investigated whether α3β1 regulates the expression of CCN2 in *Hras*-transformed keratinocytes, isolated from K14 *Itga3* WT mice that underwent the full DMBA/TPA carcinogenesis protocol (MSCC WT and MSCC *Itga3* KO keratinocytes) (Sachs et al, 2012). In agreement with our observations in mice, deletion of α3β1 in MSCC keratinocytes resulted in reduced levels of CCN2 mRNA (Fig 7A). Furthermore, MSCC WT keratinocytes showed an increased expression of CCN2 when the conditions that occur during DMBA/TPA treatment were mimicked by application of either TPA or IL-6, the cytokine up-regulated during DMBA/TPA tumorigenesis and crucial for tumor formation (Ancrile et al, 2007) (Fig 7A). Integrin α3β1-dependent expression of CCN2 and its increase upon TPA and IL-6 treatment were confirmed at the protein level using IF and Western blot (WB) analysis (Figs 7B and C and S5A). Furthermore, IF analysis showed that MSCC WT keratinocytes secrete CCN2, which co-localized with deposited laminin-332 (Figs 7B and S5B). To assess whether CCN2 contributes to the tumorigenic properties of transformed keratinocytes in vitro, we generated two CCN2 KO clones using CRISPR/Cas9 with two distinct guide RNAs (MSCC CCN2 KO G1

and MSCC CCN2 KO G2) (Fig 8A) and submitted them to colony formation assay. In line with observations in K14 *Itga3* KO mice, the deletion of α3β1 resulted in a strong reduction of colony formation and in decreased colony size (Figs 8B and S6A). Although the deletion of CCN2 did not influence colony size, the colony-forming ability of the two CCN2 KO clones was significantly reduced compared with WT MSCC and control clones (Figs 8B and S6A). Next, we tested whether secreted CCN2 can promote survival of transformed keratinocytes by treating *Itga3* KO MSCCs and CCN2 KO clones with exogenous CCN2. CCN2 treatment significantly increased the colony formation of CCN2 KO MSCCs, albeit not to the level of the MSCC control clones (Figs 8C and S6B). No differences in colony formation could be observed upon the treatment of *Itga3* KO MSCCs with CCN2 (Figs 8D and S6B), indicating that α3β1-mediated secretion of CCN2 may enhance the tumorigenic potential of keratinocytes, but is not sufficient for tumorigenesis. We further investigated whether α3β1 and CCN2 affect the 3D growth of MSCC keratinocytes using the Matrigel matrix. The results showed that at the beginning of spheroid formation, CCN2 expression was dependent on integrin α3β1 (Fig 8E). Furthermore, spheroids needed α3β1 to successfully accumulate mass in the 3D matrix (Fig 8E and F). Importantly, although CCN2 KO clones still formed 3D spheroids, their size was significantly reduced compared with CCN2 control clones and MSCC

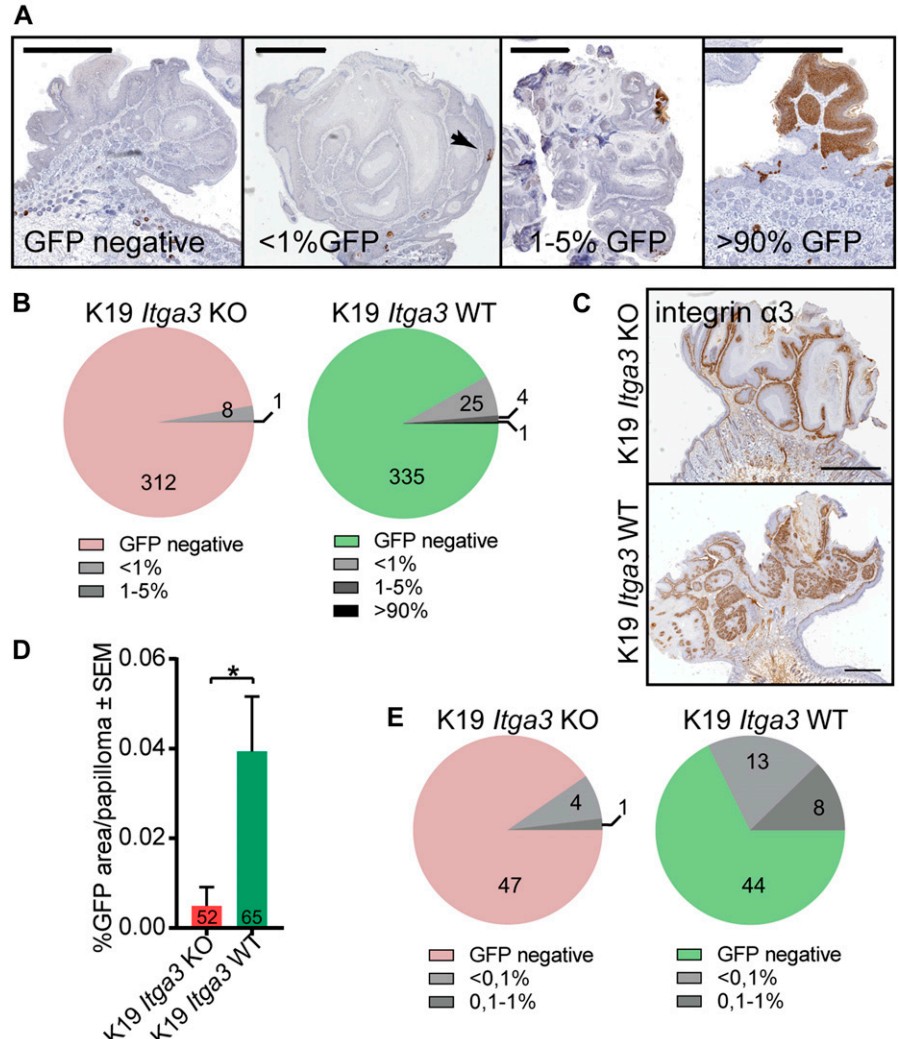

**Figure 5.  HB-derived keratinocytes are largely absent from skin tumors.**
**(A, B)** With rare exceptions, Cre-induced GFP-positive cells represent less than 1% of total tumor mass. **(A, B)** Representative IHC images stained for GFP and (B) quantification of GFP-positive area in cross sections of tumors, isolated from nine K19 *Itga3* KO and eight WT mice. The vast majority of tumors is GFP negative (scale bar: 1 mm). **(C)** Integrin α3β1 is strongly expressed in all tumors analyzed (scale bar: 1 mm). **(D, E)** Analysis of GFP-positive area over 10 cross sections, cut every 200 µm of randomly selected tumors from four K19 *Itga3* KO and four WT mice. **(D)** The contribution of GFP-positive HB-originating keratinocytes to tumors, isolated from K19 *Itga3* KO is significantly reduced compared with WT mice (mean ± SEM, unpaired *t* test, *P < 0.05). **(E)** Most tumors are GFP negative in both, K19 *Itga3* KO and WT mice. GFP was detected in 9.6% of K19 *Itga3* KO and in 32.3% of K19 *Itga3* WT tumors and did not exceed 1% of total tumor mass.

WT keratinocytes (Fig 8F). In 3D culture, seeding the CCN2 KO MSCC clones in the presence of two different concentrations of CCN2 slightly increased their growth potential (Figs 8G and S6C), which was not observed when CCN2 MSCC cells were treated 3 d after seeding, when spheroids had already been formed (Fig S6D). This is in agreement with our previous observations of low CCN2 expression during late stages of tumor and spheroid growth (Figs 8E and S4C) and could further indicate that exogenous CCN2 may enhance tumorigenic potential of transformed keratinocytes predominantly during early stages of tumorigenesis. However, as CCN2 can associate with extracellular matrix components (Ramazani et al, 2018), which we observed also in this study (Fig S5B), it is conceivable that the concentration of CCN2 that reached cells within the inner layers of spheroids was too low to promote the growth of already formed spheroids, despite the high concentration of CCN2 used (Fig S6D). Finally, we confirmed that α3β1 expression is an essential prerequisite for CCN2-mediated promotion of tumorigenesis, as seeding *Itga3* KO MSCC with CCN2 had no effect on their 3D growth (Fig 8H).

Together, these data show that CCN2 plays a role in supporting the tumorigenic potential of α3β1-expressing transformed keratinocytes in vitro, which makes it a likely player in our in vivo tumorigenesis mouse model.

## Discussion

The previous hypothesis on the mechanism behind the essential role of integrin α3β1 in DMBA/TPA–induced tumorigenesis heavily weighted on the idea that HB SCs represent the cancer cells-of-origin and that the deletion of α3β1 promotes their egress into IFE (Sachs et al, 2012). Here, we show that the deletion of α3β1 in HB keratinocytes causes a slight increase in their egress only during wound healing. As the egress of HB SCs and α3β1-mediated suppression of keratinocyte migration during wound healing is well established, such role of HB-residing α3β1 in wound re-epithelization is not unexpected (Ito et al, 2005; Margadant et al, 2009; Plikus et al, 2012). More surprising is the observation that the deletion of α3β1

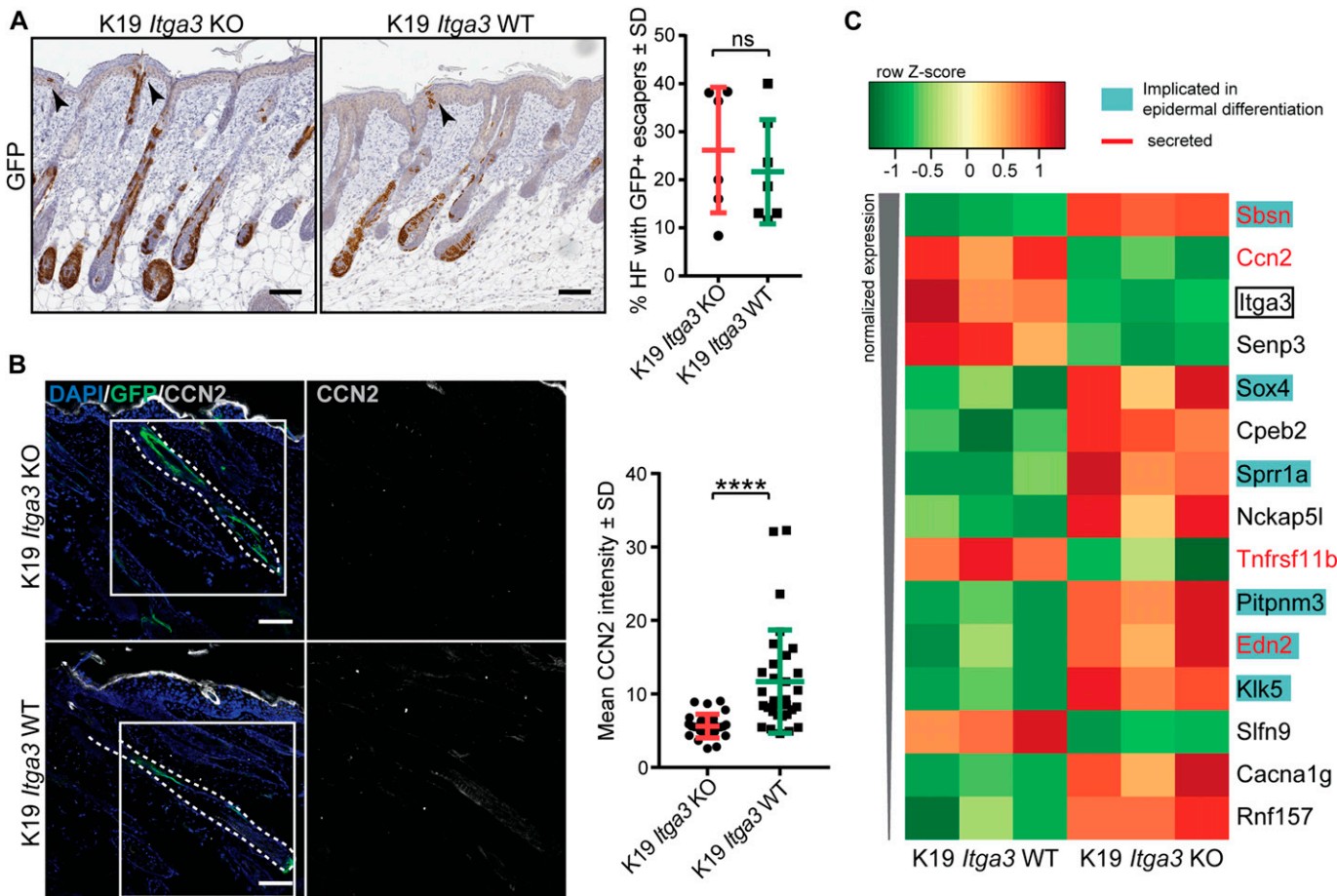

**Figure 6. α3β1-depleted keratinocytes show an increased differentiation signature and decreased expression of CCN2 during the initiation stage of tumorigenesis.**
**(A)** GFP-positive Cre-induced HB SCs localize to growing hair follicles (HFs) and, in some cases, to isthmus, infundibulum, and IFE (black arrows) after short-term DMBA/ TPA treatment in K19 *Itga3* KO and WT mice. Left: IHC staining for GFP (scale bar: 100 μm). Right: quantification of the number of HFs, where GFP-positive cells were observed in the upper parts of HFs and in adjacent IFE. Each dot represents a mouse (mean ± SD, unpaired *t* test). **(B)** Heat map of row-scaled significantly differentially expressed protein-coding genes of GFP-positive keratinocytes, isolated from three K19 *Itga3* KO and three K10 *Itga3* WT mice after short-term DMBA/TPA treatment. Protein-coding genes have an adjusted *P* < 0.05 and an average normalized expression across all samples >4 (as calculated with Voom) and a logFC > 0.6 between K19 Itga3 WT an KO mice. **(C)** IF staining (left) and quantification of mean intensity of the signal (right) for CCN2 in GFP-positive HFs after short-term DMBA/TPA treatment. Each dot represents a GFP-positive HF. HFs of five K19 *Itga3* KO and six K19 *Itga3* WT mice were quantified (mean ± SD, unpaired *t* test, *P* < 0.0001).

does not cause the egress of HB SCs under normal homeostatic conditions, even though α3β1-depleted, K15-positive keratinocytes are found in the upper parts of HF and in IFE of K14 *Itga3* KO mice (Sachs et al, 2012). As the deletion of α3β1 causes an increased epidermal turnover in K14 *Itga3* KO mice (Sachs et al, 2012), our finding that these keratinocytes express K15 de novo ties well with the reported close relationship between K15 expression and the loss of homeostasis of the epidermal differentiation program in basal-like cells (Troy et al, 2011).

The DMBA/TPA–induced skin carcinogenesis model mimics the multistage nature of cancer, in which substantial time is needed for tumors to outgrow from cancer-initiating cells. In line with this, the importance of slow-cycling, label-retaining cells in DMBA/TPA–initiated tumorigenesis has long been established (Morris et al, 1997, 2000). Therefore, a loss of initiated label-retaining SCs due to their premature terminal differentiation during increased epidermal turnover remains a plausible cause of the near complete

absence of tumorigenesis in K14 *Itga3* KO mice (Sachs et al, 2012). The finding that α3β1 affects epidermal turnover was confirmed in this study, where we observed an increased differentiation in gene signature in GFP-positive cells, isolated from K19 *Itga3* KO mice, although the differences in differentiation were not substantial enough to affect the hair cycle or hair growth in mice. This was expected, as hair cycle was not altered in K14 *Itga3* KO mice of similar age (Sachs et al, 2012), although the loss of integrin α3β1 has been shown to affect HF maintenance and morphology in mice, bred onto C57Bl/6 background (Conti et al, 2003; Sachs et al, 2012).

HB SCs have often been suggested to be the main cells-of-origin for DMBA/TPA–initiated tumors (Trempus et al, 2007; Malanchi et al, 2008; White et al, 2011; Lapouge et al, 2012). However, several studies have demonstrated that other epidermal cell populations can also act as cancer SCs in the DMBA/TPA model of skin carcinogenesis (Morris et al, 2000; Lapouge et al, 2011; van de Glind et al, 2016; Huang et al, 2017). Considering also the fact that the K15 promoter,

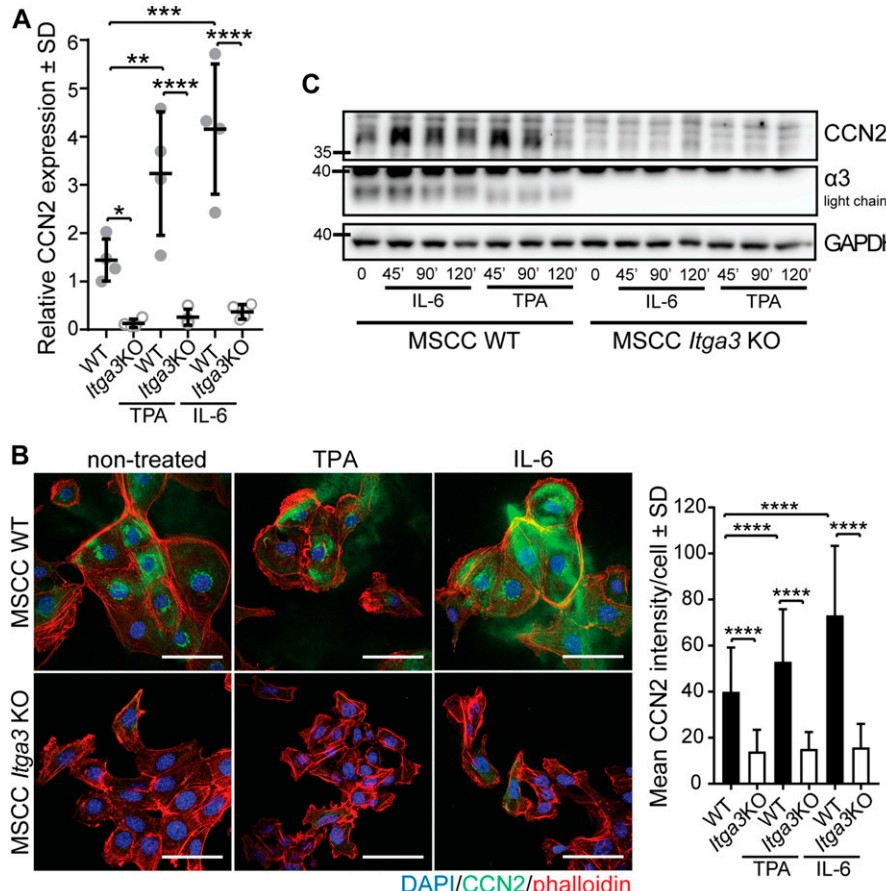

**Figure 7. CCN2 expression in transformed keratinocytes is α3β1-dependent.**
**(A)** GAPDH-normalized relative mRNA expression of CCN2 is significantly decreased in non-stimulated as well as IL-6 and TPA-treated α3β1-depleted keratinocytes. The average of up to four independent measurements of technical duplicates of four RNA samples per group (dots) is presented (mean ± SD, Fisher's LSD test, *P < 0.05, **P < 0.005, ***P < 0.0005, ****P < 0.0001). **(B)** IF (left) and quantification of the mean intensity (right) of CCN2 in non-stimulated, IL-6, and TPA-treated MSCC *Itga3* KO and WT keratinocytes. Expression of CCN2 is α3β1 dependent and increases upon IL-6 and TPA treatment (scale bar: 50 μm). 90 cells imaged over three independent experiments were quantified (mean ± SD, Fisher's LSD test, P < 0.0001). **(C)** Representative WB confirming α3β1-dependent and IL-6– and TPA-mediated CCN2 expression. Quantification can be found in Fig S5A. Source data are available for this figure.

which is commonly used for HB-specific genetic modifications, does not target this compartment exclusively (Lapouge et al, 2011), there is an emerging consensus that several epidermal SC populations can serve as cells-of-origin for DMBA/TPA–initiated tumors, albeit that there are likely differences in their fate during the progression of the disease (Morris et al, 2000; Sánchez-Danés & Blanpain, 2018). In this study, we showed that the contribution of HB SCs to skin tumors is minimal, and even further reduced in the K19 *Itga3* KO mice. A similar observation was made by Goldstein et al (2015), when they used the K19 promoter to target the deletion of the transcription factor Nfatc1 in HBs (Goldstein et al, 2015). A recent, thorough study by Reeves et al (2018) demonstrated that DMBA/TPA treatment induces papillomas that are mostly monoclonal but often contain additional minor populations at the edges of tumors. Interestingly, these minority populations, which resemble our HB-originating tumor patches, commonly do not possess driver *Hras* mutation and carry a lower mutational load (Reeves et al, 2018).

Despite their near absence in skin tumors, HB SCs affected tumor incidence and, to a much lesser degree, tumor size in an α3β1-dependent manner. The possibility that HB SCs can affect (rather than be) cancer-initiating cells has received undeservedly little attention. Only a very recent study by the research group of Valentina Greco demonstrated that HB SCs exhibit tolerance to *Hras* mutation and non-autonomously affect neighboring epithelial and

stromal cells (Pineda et al, 2019). Our observations of HB-originating GFP-positive cells in IFE and upper parts of HFs during initiation of tumorigenesis indicate that α3β1 could affect neighboring cells and/or promote a permissive tumor environment in different epidermal niches. Furthermore, small HB-derived cell populations in tumors may also affect the survival and/or proliferation of neighboring tumor cells.

It is well established that the cellular environment, composed of fibroblasts, endothelial cells, and immune cells, plays a crucial role in the development and progression of DMBA/TPA–initiated tumors (Zhang et al, 2011; Lapouge et al, 2012; Medler & Coussens, 2014; Neagu et al, 2016). Over the past few years, the role of integrins beyond adhesion and biomechanical transduction of signals has become more evident, especially in their ability to modulate the tumor microenvironment (Longmate & DiPersio, 2017). In particular, integrin α3β1 has been shown to regulate paracrine signaling and cross talk between cells, among others by controlling the expression of secreted cellular proteins (Mitchell et al, 2010; Longmate et al, 2017; Zheng et al, 2019). This is in line with our findings that the expression of CCN2 in HB keratinocytes in vivo and in mouse keratinocytes in vitro depends on α3β1. CCN2 has been implemented in skin tumorigenesis previously and is an interesting potential player in DMBA/TPA–induced skin cancer because its expression can be regulated by several proteins, essential for tumor formation in this model, such as FAK, YAP, TGFβ, and Stat3

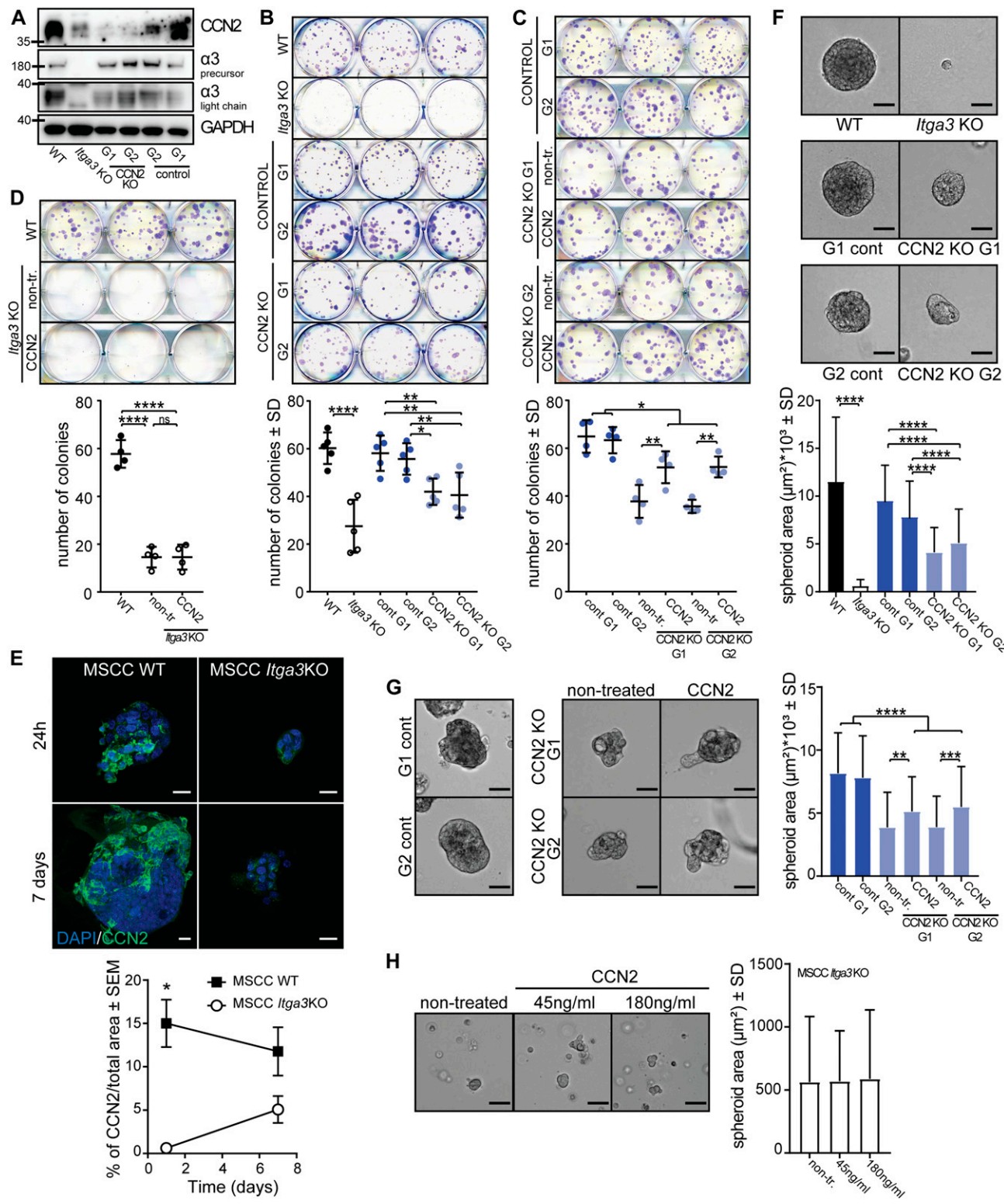

**Figure 8. CCN2 promotes colony formation and 3D growth of transformed keratinocytes expressing α3β1.**
**(A)** WB of CCN2 and integrin α3-expression of selected CCN2 KO and control clones. **(B)** Representative image (top) and quantification (bottom) of colony-formation assay of MSCC *Itga3* WT and *Itga3* KO cells and MSCC CCN2 KO G1, KO G2, control G1, and control G2 clones. Deletion of α3β1 results in a strong reduction of colony formation and colony size. Moderate reduction of colony formation can be seen upon CCN2 deletion. Quantification of colony size can be found in Fig S6A. Average values of technical triplicates of five independent experiments are presented (mean ± SD, Fisher's LSD test, *P < 0.05, **P < 0.005, ****P < 0.0001). **(C)** Representative image (top) and quantification (bottom) of colony-formation assay of MSCC CCN2 KO G1 and KO G2 clones, grown in control conditions or in the presence of 45 ng/ml CCN2, as well as CCN2

(McLean et al, 2004; Kim et al, 2009; Pérez-Lorenzo et al, 2010; Quan et al, 2014; Zanconato et al, 2015; Ramazani et al, 2018). Moreover, it can directly interact with integrins, as well as with cytokines, such as TGFβ, and heparan sulfate proteoglycans, which are abundantly present in the basement membrane of the skin, and thus can modulate related signal transduction pathways and their cross talk (Ramazani et al, 2018). Furthermore, it was shown that the expression of CCN2 in mesenchymal cells in the skin and lungs affects a differentiation program of adjacent epithelial cells (Sonnylal et al, 2013), which fits well with the role of α3β1 in the DMBA/TPA tumorigenesis model. As the expression of CCN2 in full-grown papillomas is very low and could not be correlated with HB-originating areas by histochemistry, its potential pro-tumorigenic role likely occurs during the initiation stage of DMBA/TPA–driven tumorigenesis. This is supported by our in vitro studies, where CCN2 increased the clonogenic potential and initiation of 3D growth of transformed keratinocytes. However, it is clear that these in vitro studies cannot be used for drawing definitive conclusions concerning the role of CCN2 in vivo, and that further work is needed to explore this role. It would be particularly interesting to see if the deletion of CCN2 in HBs reduces the formation of papillomas in mice subjected to the two-stage carcinogenesis protocol.

It is important to note that the effect of CCN2 deletion on the in vitro tumorigenic properties of transformed keratinocytes is weak to moderate compared with the pronounced effect of the deletion of *Itga3*. Furthermore, the ability of CCN2 to promote the survival and outgrowth of transformed keratinocytes depends on the expression of α3β1. Thus, CCN2 may enhance tumorigenesis but is not essential for it. Considering also the fact that the deletion of α3β1 in HB SCs fails to completely recapitulate the effect of total epidermal α3β1 deletion on tumorigenesis, there must be additional α3β1-dependent functions that are relevant to tumor initiation and formation. Therefore, our study reopens the question of the mechanism behind the essential role of α3β1 in DMBA/TPA–driven tumorigenesis. Here, we disprove the original hypothesis that the dramatic effect of epidermal α3 deletion can be explained by the egress of HB SCs. Furthermore, we show that α3β1 in HB SCs contributes to tumorigenesis; however, its moderate effect strongly indicates that there are additional mechanisms in play, which will have to be re-examined.

Even though K19 promoter targets the HB with high specificity, a limitation of the K19–CreER mouse model is the low efficiency of Cre-mediated recombination. We have observed Cre induction in most HFs; however, α3β1 has not been deleted in all the HBs, which could mean that the contribution of HB-originating cells to tumor mass has been understated. With this concern in mind, we have analyzed a large number of tumors for HB originating, GFP-positive cells. As we found only 1 tumor of 365 to consist entirely of GFP-positive cells, we believe that the conclusion that HB SCs do not represent the main cancer cells-of-origin is credible. The remaining α3β1-positive cells in HBs of K19 *Itga3* KO mice could also lead to skewed effects of the deletion of α3β1 in HBs in tumorigenesis assays. This is of lesser concern, as we observed non-efficient deletion of α3 also in the HBs of K14 *Itga3* KO mice, that is, the model in which DMBA/TPA treatment resulted in near absence of tumor formation. Furthermore, it should be noted that differences between the two models may arise because of the different timing of *Itga3* deletion in the K14 *Itga3* KO and K19 *Itga3* KO mice. However, the subtility of the phenotype of constitutive *Itga3* epidermal deletion at the age when HB-specific *Itga3* deletion was induced in K19-driven mouse model offers some reassurance that our findings with the K19 *Itga3* KO mice can be related to those with the K14 *Itga3* KO mice (Margadant et al, 2009).

In conclusion, we show that α3β1 in HB SC population indirectly contributes to skin tumorigenesis. Integrin α3β1-regulated expression of matricellular protein CCN2 during the initiation of tumorigenesis indicates that α3β1 might mediate paracrine signaling and, thus, promote the formation of a permissive tumor environment. The role of CCN2 as a potential player in α3β1-mediated tumorigenesis was demonstrated in transformed keratinocytes in vitro. Although our findings elucidate only parts of the mechanism underlying the essential function of α3β1 in DMBA/TPA–driven tumorigenesis, they provide a new understanding of the role of HB SCs in skin tumorigenesis as well as offer an important insight into complex and diverse ways in which integrins can affect this disease.

# Materials and Methods

### Generation of mice

K19-CreER mice (Means et al, 2008) were intercrossed with mT/mG mice (Gt(ROSA)26Sor^tm4(ACTB-tdTomato,−EGFP)Luo, strain 007576; Jackson Laboratory) to obtain K19-CreER; mT/mG mice (K19 *Itga3* WT mice). K19-CreER; mT/mG mice were further intercrossed with *Itga3*^fl/fl (i.e., *Itga3*^tm1Son/tm1Son according to the Mouse Genome Informatics) to

---

WT control G1 and control G2 clones. Treatment with exogenous CCN2 significantly increases colony formation of CCN2 KO clones. Quantification of colony size can be found in Fig S6B. Average values of technical triplicates of four independent experiments are presented (mean ± SD, Fisher's LSD test, *P < 0.05, **P < 0.005). **(D)** No differences in the number of colonies can be observed upon CCN2 treatment of *Itga3* KO–transformed keratinocytes. Quantification of colony size can be found in Fig S6B. Average values of technical triplicates of four independent experiments are presented (mean ± SD, Fisher's LSD test, ****P < 0.0001). **(E)** Quantification (left) and IF as maximum intensity projection (right) of CCN2 expression of MSCC *Itga3* WT and KO spheroids, grown in 3D Matrigel matrix for 1 or 7 d (scale bar: 20 μm). The expression of CCN2 is α3β1 dependent, which is particularly prominent at the beginning of spheroid growth. The percentage of CCN2-positive area was quantified from 17 MSCC *Itga3* KO and 30 MSCC *Itga3* WT spheroids 1 d after seeding and from 15 MSCC *Itga3* KO and 27 MSSC WT spheroids 7 d after seeding (mean ± SEM, unpaired t test, *P < 0.05). **(F)** Spheroid growth in 3D Matrigel is α3β1 dependent and moderately reduced upon CCN2 deletion. Top: bright-filed images of representative spheroids (scale bar: 50 μm). Bottom: size quantification of 60–80 spheroids measured over three to four independent experiments (mean ± SD, Fisher's LSD test, ****P < 0.0001). **(G)** 3D growth of CCN2 KO MSCC spheroids shows small but significant increase when cells are seeded with 45 ng/ml of CCN2. Left: bright-filed images of representative spheroids (scale bar: 50 μm). Right: size quantification of 85–90 spheroids measured over three independent experiments (mean ± SD, Fisher's LSD test, **P < 0.005, ***P < 0.0005 ****P < 0.0001). **(H)** Seeding *Itga3* KO MSCCs with 45 or 180 ng/ml of CCN2 does not impact the 3D growth pf spheroids. Left: bright-filed images of representative spheroids (scale bar: 50 μm). Right: size quantification of 70 spheroids measured over two independent experiments (mean ± SD, one-way ANOVA, P = 0.9491). Source data are available for this figure.

obtain K19-CreER; mT/mG; _Itga3_<sup>fl/fl</sup> (K19 _Itga3_ KO) mice. Epidermis-specific mice are named Krt14<sup>tm1(cre)Wbm</sup> (K14 _Itga3_ WT mice) and Krt14<sup>tm1(cre)Wbm</sup>; _Itga3_<sup>tm1Son/tm1Son</sup> (K14 _Itga3_ KO mice) according to the Mouse genome Informatics and have been described before (Sachs et al, 2012). All mice were bred onto an FVB/N background. All animal studies were performed according to the Dutch guidelines for care and use of laboratory animals and were approved by the animal welfare committee of the Netherlands Cancer Institute.

## Animal experiments

A DMBA/TPA carcinogenesis protocol has been previously described (Sachs et al, 2012). Briefly, backs of 6-wk-old mice were shaved and after a week topically treated with 30 µg (in 200 µl acetone) of DMBA (D3254; Sigma-Aldrich), followed by bi-weekly topical applications of 12.34 µg (in 200 µl acetone) of TPA (P1585; Sigma-Aldrich) for 20 wk. A similar procedure was used for short-term DMBA/TPA treatment; however, mice were only treated with four doses of TPA over 2 wk after DMBA treatment. To induce Cre-recombinase, 5-wk and 5-d-old mice were injected IP for 4 d with 2.5 mg/ml of tamoxifen (T5648; Sigma-Aldrich) dissolved in sunflower oil per day and (at 6 wk) additionally topically treated with two doses of 200 µl of 20 mg/ml of tamoxifen dissolved in ethanol. For linage tracing, 3-wk-old mice were given an intraperitoneal injection of 2.5 mg/ml of tamoxifen dissolved in sunflower oil per day for four consecutive days. After indicated time points, mice were killed, and the skin was isolated and processed for immunofluorescence, immunohistochemistry or flow cytometry analysis, and/or for RNA isolation. Wound-healing experiments were performed as described previously (Ketema et al, 2015). 5-wk and 5-d-old mice were given an intraperitoneal injection of 2.5 mg/ml of tamoxifen dissolved in sunflower oil per day for four consecutive days. 7-wk-old mice were anesthetized and shaved, and four full-thickness excision wounds of 4-mm diameter were cut with small scissors (two per either side of the dorsal midline). Complete wounds, including surrounding tissue, were excised 3 and 5 d after injury. Paraffin

sections across the middle of the wounds were used for histological analysis of the wound closure. 10 consecutive sections cut every 100 µm were used for quantification of the GFP-positive area per area of neo-epidermis.

## Immunohistochemistry

Skin, tumors, and tails were isolated, fixed in ethanol glacial acetic acid mixture (3:1), containing 2% of formaldehyde (EAF) and/or formaldehyde, embedded in paraffin, sectioned, and stained for hematoxylin and eosin (H&E) and/or immunohistochemistry (see Table 1). Images were taken with PL APO objectives (10×/0.25 NA, 40×/0.95 NA, and 63×/1.4 NA oil) on an Axiovert S100/AxioCam HR color system using AxioVision 4 software (Carl Zeiss MicroImaging) or with the Aperio ScanScope (Aperio), using ImageScope software version 12.0.0 (Aperio). Tumor classification was performed blindly. Image analysis was performed using ImageJ (Schindelin et al, 2012; Rueden et al, 2017).

## Immunofluorescence and whole mounts

Skin was isolated and embedded in Tissue-Tek OCT (optimal cutting temperature) cryoprotectant. Cryosections of skin were prepared, fixed in ice-cold acetone, and blocked with 2% BSA (Sigma-Aldrich) in PBS for 1 h at room temperature. Whole mounts of tail epidermis were isolated as described previously (Sachs et al, 2012), fixed in 4% paraformaldehyde in PBS, and permeabilized and blocked in PB buffer (20 mM Hepes buffer, pH 7.2, containing 0.5% [vol/vol] TrotonX-100, 0.5% [wt/vol] skim milk powder, and 0.25% [vol/vol] fish skin gelatin). MSCC cells were fixed with 2% paraformaldehyde for 10 min, permeabilized with 0.2% Triton X-100 for 5 min, and blocked with PBS containing 2% BSA for 1 h at room temperature. Spheroids were retrieved from Matrigel by incubation with Cell Recovery Solution (354253; Corning) for 1 h at 4°C and subsequently resuspended in ice-cold PBS. Isolated spheroids were mounted on poly-L-lysine (25988-63-0; Santa Cruz)–coated slides, fixed in 4% paraformaldehyde in PBS for 10 min, permeabilized with 0.2% Triton

**Table 1. List of primary antibodies used, including application, dilution, and source.**

| Antigen | Name | Type | Application | Dilution | Source |
|---|---|---|---|---|---|
| Integrin α3 | | Rabbit pAb | WB | 1:2,000 | Homemade |
| Integrin α3 | AF2787 | Goat pAb | IF | 1:100 | R&D Systems |
| Integrin α3 | AF2787 | Goat pAb | FACS | 1:100 | R&D Systems |
| Integrin α3 | sc-374242 | Mouse mAb | IHC | 1:500 | Santa Cruz |
| Integrin α6-PE | eBioGoH3 | Rat mAb | FACS | 1:200 | eBioscience |
| CCN2 | E-5 | Mouse mAb | WB | 1:800 | Santa Cruz |
| CCN2 | L-20 | Goat pAb | IF, IHC | 1:100 | Santa Cruz |
| CD34-FITC | RAM34 | Rat mAb | FACS | 1:100 | eBioscience |
| GAPDH | CB1001 | Mouse mAb | WB | 1:1,000 | Calbiochem |
| GFP | ab6556 | Rabbit pAb | IHC | 1:2,000 | Abcam |
| Keratin 15 | MA1-90929 | Mouse mAb | IF, IHC | 1:200 | Thermo Fisher Scientific |
| Ki67 | PSX1028 | Rabbit pAb | IHC | 1:750 | Monosan |
| Laminin-332 | R14 | Rabbit pAb | IF | 1:400 | Kind gift of M Aumailey |

X-100 for 5 min, and blocked with PBS-containing 2% BSA for 1 h at room temperature. Tissues, spheroids, and cells were incubated with the indicated primary antibodies (see Table 1) in 2% BSA in PBS (whole mounts: PB buffer) for 60 min (whole mounts and spheroids: overnight), followed by incubation with secondary antibodies diluted 1:200 for 60 min (whole mounts: overnight). All samples where counterstained with DAPI for 5 min at room temperature and, when indicated, filamentous actin was visualized using Alexa Fluor 488–conjugated phalloidin (Invitrogen). Cryosections were mounted in Vectashield (H-1000; Vector Laboratories) and other samples in Mowiol. Samples were analyzed by Leica TCS SP5 confocal microscope with a 10 or 20× (NA 1.4) objective or 40 and 63× (NA 1.4) oil objective and processed using ImageJ (Schindelin et al, 2012; Rueden et al, 2017).

### Flow cytometry

Keratinocytes were isolated from mouse back skin as described before (Jensen et al, 2010), washed in PBS containing 2% FCS, and incubated for 1 h at 4°C in primary antibody (see Table 1) in PBS 2% FCS. In case of non-fluorophore–conjugated antibodies, the cells were subsequently incubated with donkey antigoat Alexa Fluor 647 (1:200 dilution; Invitrogen) antibody for 30 min at 4°C. The cells were analyzed on a Becton Dickinson FACSCalibur analyzer after the addition of indicated life/dead cell marker. For fluorescent-activated cell sorting, GFP-positive cell population was obtained using a Becton Dickinson FACSAria IIu cell sorter.

### RT-qPCR

Keratinocytes were isolated from mouse back skin as described before (Jensen et al, 2010), pelleted by centrifugation, and resuspended in TRIzol Reagent (15596018; Invitrogen). Whole mounts of tail epidermis were isolated (Sachs et al, 2012) and homogenized in TRIzol Reagent using Polytron, while keeping the temperature of the tissue at 4°C. 10-cm dishes of semi-confluent cells were washed with ice-cold PBS, scraped, and collected in 2 ml of TRIzol Reagent. Total RNA was extracted according to the manufacturer's recommendations (TRIzol Reagent), and 3 μg of purified RNA was used to synthesize the first-strand cDNA using First-Strand cDNA Synthesis Kit (K1612; Thermo Fisher Scientific). Quantitative PCR analyses were performed using a Sybr Green gPCR Master Mix (K0251; Thermo Fisher Scientific) and ABI Prism 7500 Real-Time PCR System (Applied Biosystems). Analysis of results was performed with 7500 Fast System SDS software v 1.4 (Applied Biosystems). Results were presented as relative quantification of the ratio of K15 or CCN2 Ct over the GAPDH Ct. Following primers were used 5′-TGA-GAAGGTGACCATGCAGA-3′ and 5′-GGCAGCCAGAATCGGATCTC-3′ for keratin 15, 5′-AGAACTGTGTACGGAGCGTG-3′ and 5′-GTGCACCATCTTTGGCAGTG-3′ for CCN2 and 5′-ACTCCACTCACGGCAAATTC-3′ and 5′-TCTCCATGGTGGT-GAAGA-3′ for GAPDH.

### Expression analysis of HB-originating keratinocytes

Keratinocytes from the back skin of short-term DMBA/TPA–treated mice were isolated as described before and FACS-sorted for GFP-positive cells. Total RNA was isolated using the RNeasy Mini Kit (74106; QIAGEN), including an on-column DNase digestion (79254;

QIAGEN), according to the manufacturer's instructions. Quality and quantity of the total RNA was assessed by the 2100 Bioanalyzer using a Nano chip (Agilent). Total RNA samples having an RNA integrity number (RIN) >8 were subjected to library generation. Strand-specific libraries were generated using the TruSeq Stranded mRNA Sample Preparation Kit (RS-122-2101/2; Illumina Inc.) according to the manufacturer's instructions (part no. 15031047 Rev. E; Illumina Inc.). Briefly, polyadenylated RNA from intact total RNA was purified using oligo-dT beads. After purification, the RNA was fragmented, random-primed, and reverse-transcribed using SuperScript II Reverse Transcriptase (part no. 18064-014; Invitrogen) with the addition of actinomycin D. Second strand synthesis was performed using polymerase I and RNaseH with replacement of deoxythymidine triphosphate (dTTP) for deoxyuridine triphosphate (dUTP). The generated cDNA fragments were 3′ end adenylated and ligated to Illumina paired-end sequencing adapters and subsequently amplified by 12 cycles of PCR. The libraries were analyzed on a 2100 Bioanalyzer using a 7500 chip (Agilent), diluted, and pooled equimolar into a multiplex sequencing pool. The libraries were sequenced with 65 base single reads on a HiSeq2500 using V4 chemistry (Illumina Inc.). Demultiplexing of the reads was performed with Illumina's bcl2fastq. Demultiplexed reads were aligned against the mouse reference genome (build 38) using TopHat (version 2.1.0, bowtie 1.1). TopHat was supplied with a known set of gene models (Gene Transfer Format file Ensembl version 77) and was guided to use the first strand as the library type. Prefilter multihits and no coverage were used as additional parameters. To count the number of reads per gene, a custom script which is based on the same ideas as HTSeq count has been used. For each sample, uniquely mapped sequencing reads were used to get the gene counts for all genes present in the Gene Transfer Format file. The strandedness of the mapped reads was taken into account. Differentially expressed analysis was performed using the R packages Limma/EdgeR. Reads that have zero counts across all samples were removed from the dataset. The samples were normalized to counts per million using the "Voom" function from the Limma package. Gene expressions used for visualization purposes are normalized to 10 million reads and $\log_2$ transformed. Before $\log_2$ transformation, one pseudocount was added to avoid negative gene expressions.

### Cell culture

MSCC WT and *Itga3* KO cells were generated as described (Sachs et al, 2012) and cultured in DMEM with 10% heat-inactivated FCS and antibiotics at 37°C in a humidified, 5% $CO_2$ atmosphere. For CRISPR/Cas9–mediated CCN2 deletion, we cloned target single guide (sg) RNAs against CCN2 (exon 2; guide 1: 5′-ACTCCGATCTTGCGGTTGGC-3′; guide 2: 5′-CTCCGATCTTGCGGTTGGCG-3′) into pX330.pgkpur vector (a kind gift from the laboratory of Hein te Riele [Harmsen et al, 2018]). MSCC WT cells were transiently transfected with this vector using Lipofectamine 2000 (Invitrogen). The cells were selected with 5 μg/ml of puromycin, and puromycin-resistant bulk population was single-cell cloned. The clones were analyzed for the expression of CCN2 using WB analysis. For IL-6 and TPA stimulation of MSCC, we added 10 ng/ml of recombinant human IL-6 (206-IL; R&D Systems) or 100 ng/ml of TPA to DMEM 10% FCS and incubated cells with the mixture for 45 min (for IF and RT-qPCR analysis) or for 45, 90, and 120

min (for WB analysis). For 3D cell culture, 70 $\mu$l of growth factor reduced Matrigel Basement Membrane Matrix (354230; Corning) was pipetted per well of chilled 96-well plate and incubated for 30 min at 37°C. 1,000 cells in cold DMEM containing 10% FCS and 2% Matrigel were seeded on top of Matrigel layer and grown for up to 7 d. For CCN2 treatment experiments, the cells were seeded in the presence of 45 or 180 ng/ml of mouse recombinant CCN2 (RD272589025; Biovendor) and cultured for up to 9 d. Where indicated, 180 ng/ml of CCN2 was added to growth medium of non-treated spheroids on day 3. Medium and CCN2 was refreshed every 3 d of 3D cell culture.

### Colony-formation assay

100 cells in DMEM 10% FCS were seeded per well of a six-well plate and incubated at 37°C in a humidified, 5% $CO_2$ atmosphere for 6 d. Where indicated, 45 ng/ml of CCN2 was added to cells during seeding. Colonies were fixed in chilled methanol for 20 min, washed with PBS, and stained with crystal violet. Stained plates were scanned and colony number and size was quantified using ImageJ (Schindelin et al, 2012; Rueden et al, 2017).

### WB

Protein lysates were obtained from subconfluent cell cultures by lysis in RIPA buffer (20 mM Tris–HCl, pH 7.5, 100 mM NaCl, 4 mM EDTA, pH 7.5, 1% Nonidet P-40, 0.5% sodium deoxycholate, and 0.1% SD) supplemented with 1.5 mM $Na_3VO_4$, 15 mM NaF, and protease inhibitor cocktail (Sigma-Aldrich). Lysates were cleared by centrifugation at 14,000$g$ for 20 min at 4°C and diluted in sample buffer (50 mM Tris–HCl, pH 6.8, 2% SDS, 10% glycerol, 12.5 mM EDTA, and 0.02% bromophenol blue) with final concentration of 2% $\beta$-mercaptoethanol and denatured at 95°C for 10 min. Proteins were separated by electrophoresis using Bolt Novex 4–12% gradient Bis-Tris gels (Invitrogen), transferred to Immobilon-P transfer membranes (Millipore Corp), and blocked for 1 h in 2% BSA in TBST buffer (20 mM Tris, pH 7.5, 150 mM NaCl, and 0.2% Tween-20). The blocked membranes were incubated overnight at 4°C with primary antibodies (see Table 1) in TBST containing 2% BSA, following by 1-h incubation at room temperature with horseradish peroxidase–conjugated goat-antimouse IgG or goat-antirabbit IgG (diluted 1:5,000 in 2% BSA in TBST buffer). After washing, bound antibodies were detected by enhanced chemiluminescence using or Clarity Western ECL Substrate (Bio-Rad), and signal intensities were quantified using ImageJ (Schindelin et al, 2012; Rueden et al, 2017).

### Antibodies

Primary antibodies used are listed in Table 1. Secondary antibodies were donkey-antirabbit Alexa 594, donkeyantigoat Alexa 594, donkey-antigoat Alexa 647, goat-antimouse Alexa 647 (Invitrogen), stabilized goat-antimouse HRP-conjugated, and stabilized goat-antirabbit HRP-conjugated (Bio-Rad).

### Statistical analysis

Statistical analysis was performed using GraphPad Prism (version 7.0c). Graphs represent the mean and error bars SD or SEM, as indicated per graph. Unpaired two-tailed $t$ test was used for comparisons of experimental groups with a control group. One-way ANOVA was used to analyze experiments with more than two groups and two-way ANOVA was performed on experimental data where we investigated the effect of different treatments on two distinct cell types. Type I errors were reduced by testing only planned comparisons among a relatively small number of means. Planned comparisons were conducted using Fisher's least significant difference test after a global ANOVA was determined to be significant. The significant values shown are described in appropriate figure legends. Results with $P$-value lower than 0.05 were considered significantly different from the null hypothesis.

## Data Availability

The RNA sequencing data from this publication have been deposited to the GEO database and assigned the identifier GSE135983.

## Supplementary Information

## Acknowledgements

We would like to acknowledge Alba Zuidema for the help with mouse work and for helpful discussions. We further would like to thank Reinhard Fässler for critical reading of the manuscript and Guoqiang Gu for sharing K19-CreER mice. This work was supported by a grant from the Dutch Cancer Society (project 5971).

### Author Contributions

V Ramovs: conceptualization, data curation, formal analysis, validation, investigation, visualization, methodology, and writing—original draft.
A Krotenberg Garcia: investigation and visualization.
J-Y Song: formal analysis, validation, investigation, and visualization.
I de Rink: formal analysis, validation, visualization, and methodology.
M Kreft: methodology.
R Goldschmeding: resources.
A Sonnenberg: conceptualization, supervision, funding acquisition, validation, visualization, methodology, and writing—review and editing.

### Conflict of Interest Statement

The authors declare that they have no conflict of interest.

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
