## [Reviewer comments · Life Science Alliance]

Life Science Alliance

Integrin $\alpha 3\beta 1$ in hair bulge stem cells modulates CCN2 expression and promotes skin tumorigenesis

Veronika Ramovs, Ana Krotenberg Garcia, Ji-Ying Song, Iris de Rink, Maaïke Kreft, Roel Goldschmeding, and Arnoud Sonnenberg

DOI: <https://doi.org/10.26508/lsa.202000645>

Corresponding author(s): Arnoud Sonnenberg, Netherlands Cancer Institute

Review Timeline:

Submission Date:	2020-01-13
Editorial Decision:	2020-02-10
Revision Received:	2020-04-09
Editorial Decision:	2020-04-30
Revision Received:	2020-05-06
Accepted:	2020-05-06

Scientific Editor: Andrea Leibfried

Transaction Report:

February 10, 2020

Re: Life Science Alliance manuscript #LSA-2020-00645

Prof. Arnoud Sonnenberg
Netherlands Cancer Institute
Division of Cell Biology
Plesmanlaan 121
Amsterdam 1066 CX
Netherlands

Dear Dr. Sonnenberg,

Thank you for submitting your manuscript entitled "Integrin $\alpha 3\beta 1$ in hair bulges promotes skin tumorigenesis through modulation of microenvironment" to Life Science Alliance. The manuscript was assessed by expert reviewers, whose comments are appended to this letter.

As you will see, the reviewers appreciate your analyses and they support further consideration here. We would thus like to invite you to submit a revised version of your work, addressing the specific concerns of all reviewers. This seems feasible and rather straightforward, but please do get in touch in case you would like to discuss individual points further with us. Rev#3 provides constructive input on how to better support the proposed link of CCN2 to $\alpha 3\beta 1$ -dependent effects on tumor initiation and/or growth, and we think it would be important to do so.

Thank you for this interesting contribution to Life Science Alliance. We are looking forward to receiving your revised manuscript.

Sincerely,

B. MANUSCRIPT ORGANIZATION AND FORMATTING:

Reviewer #1 (Comments to the Authors (Required)):

This manuscript addresses the cellular origin of skin cancers, which is an important and somewhat controversial question. Deletion of $\alpha3\beta1$ in keratinocytes decreases DMBA/TPA tumorigenesis. Previous work from the same group suggested that the cause of the decrease was differentiation

and movement of hair follicle bulge cells out of their niche. In this manuscript, the investigators delete $\alpha3\beta1$ in the bulge using a K19 driven Cre. They find that tumors still decrease but that bulge cells do not exit their niche. Using RNA seq they find CCN2 decreased in the $\alpha3\beta1$ deleted cells and propose that $\alpha3\beta1$ in the bulge cells controls tumor formation through this factor and the secretome in general. The findings are presented clearly and in a balanced manner. The limitations of the study are outlined in the discussion. Overall the manuscript adds to our understanding of bulge cells in tumor formation in this model.

A few minor points:

1. The authors correctly point out that two-stage carcinogenesis studies on CCN2 deleted mice would provide the strongest evidence to support their hypothesis. The paper would be much stronger if they had these data. However, the manuscript in its current form presents enough data in a balanced manner that supports the notion and will spur the additional studies along these lines.
2. The authors state that adult hairless mice lack bulge cells. This is not the case. K15 positive bulge cells are still present in the hair follicles of hairless mice (see Miller et al, JID, 2001)
3. The authors state that bulge derived keratinocytes do not permanently populate IFE. This is somewhat controversial as Sonny Wang at Univ Michigan found long-lived keratinocytes in the IFE apparently derived from bulge cells. It is worth mentioning his study.

Reviewer #2 (Comments to the Authors (Required)):

This paper by Ramovs et al. presents evidence that $\alpha1\beta1$ in the hair follicle bulge indirectly contributes to skin carcinogenesis through changes in signaling leading to a permissive tumor microenvironment. The authors identify CCN2 as a potential target for this process by its deletion leading to reduced growth and survival of keratinocytes in vitro. These new results provide increased understanding of the role of hair follicle bulge stem cells in skin carcinogenesis and offers insight into the action of integrins in this process. This paper makes interesting claims that are significant, and should interest researchers in hair follicle stem cell biology and cutaneous carcinogenesis.

There are two weaknesses. First, the paper falls short of demonstrating a mechanism for the role of $\alpha3\beta1$ in cutaneous carcinogenesis. I would be desirable if the authors could provide additional data supporting the mechanism of modulation of the bulge stem cell secretome. Second, some moderate English language editing would also be helpful.

General observations

1. The Abstract provides a succinct summary of the work presented.
2. The Introduction provides an effective literature review to serve as background for the studies presented.
3. The experimental Results support the general conclusion that $\alpha3\beta1$ integrin in stem cells of the hair follicle bulge promote formation of a tumor permissive environment by modulating their secretome but falls short of detailing a mechanism.
4. The Methods detail sufficient information that would enable another lab to replicate the results presented here.
5. The photomicrographs are of good quality. The Figures are adequately labeled. The sample sizes and statistical analyses are adequate.
6. The supplemental data include necessary controls and supporting data for the Results.

7. The Discussion generally sets the Results in the context of past and present literature and observations from other laboratories.

Specific Comments

1. The rationale for the "brief" TPA treatment is not clear. In light of the papers of Darwich, Glick, and Yuspa, the brief TPA treatment should not be construed as a shortcut for the 20-week promotion regimen. These investigators discovered that a brief course of TPA produced fewer tumors than the 20-week treatment but nearly all of them were "high-risk" and underwent malignant conversion whereas the 20-week promotion groups had many papillomas but few carcinomas. These observations were corroborated by RNA microarrays that, in the brief promotion groups, an expression signature of inflammation was produced. Consideration of the Darwich work should be considered in the Discussion, and a rationale given in the Results.
2. In light of the authors' evidence for tumor initiating cells outside the hair follicle bulge, the work of Morris et al. (2000, Cancer Research) should also be considered. Using a very precise epidermal abrasion protocol, these investigators reported that the targets of carcinogen and promoter action are found in the interfollicular epidermis or follicular infundibulum as well as in the hair follicles. Specifically, they found that the abraded mice developed approximately half the number of papillomas than their unabraded counterparts, but that small differences in the number of carcinomas were not statistically supported.

Reviewer #3 (Comments to the Authors (Required)):

In a previous PNAS paper this group discovered an essential role for integrin $\alpha3\beta1$ in skin tumorigenesis, using a model of K14-driven Cre to constitutively delete *Itga3* from the interfollicular epidermis (IFE) and hair follicles (Sachs et al., 2012). In the current study they use a K19-driven, tamoxifen-inducible Cre to delete *Itga3* in hair bulge stem cells (HB SCs), allowing them to delve more deeply into roles that $\alpha3\beta1$ may play from within this niche. The experiments are well performed and technically sound, and this group has extensive experience investigating $\alpha3\beta1$ functions in wound healing and cancer. Interestingly, the findings suggest that $\alpha3\beta1$ in the HB niche is less important than previously thought and are consistent with the idea that HB SCs are not the main reservoir of tumor initiating cells in the DMBA/TPA model. Moreover, the authors show that SCs remain confined to the HB regardless of $\alpha3\beta1$ expression, prompting them to alter their previous hypothesis that absence of $\alpha3\beta1$ leads to efflux of SCs from the HB into the IFE (Sachs et al., 2012). This previous conclusion was drawn from results in the K14-Cre $\alpha3$ KO model, wherein K15 (an SC marker) was detected in the IFE of $\alpha3\beta1$ -deficient epidermis. Their current findings in the K19-Cre $\alpha3$ KO model support an alternative explanation that K15 is turned on de novo when $\alpha3\beta1$ is deleted from keratinocytes of the upper HF and IFE using K14-Cre. Thus, a clear goal of this study is to re-evaluate some of the conclusions drawn from the previous PNAS paper, which they explain "reopens the question of mechanism". This is a laudable goal, and reinterpretation of the previous data in light of interesting and unexpected new findings is important to report. There is also novelty in the findings that implicate connective tissue growth factor (CCN2/CTGF) as a potential player in $\alpha3\beta1$ -dependent tumorigenesis. However, the presentation is often confusing in its attempt to reconcile current and past observations in the two genetic models, and apparent inconsistencies make it difficult to follow the authors' thinking. These aspects can be clarified in the writing (see specific comments below). The study also falls short of robustly supporting a direct role for CCN2 in $\alpha3\beta1$ -dependent tumorigenesis. While testing a direct role for CCN2 in tumorigenesis in vivo is beyond the scope of this study, there are a few straightforward experiments that could be done with existing materials and reagents to reinforce the proposed link of CCN2 to $\alpha3\beta1$ -

dependent effects on tumor initiation and/or growth (see below).

Specific comments:

1. Throughout the paper there seems to be a struggle to reconcile differences that this group has seen in DMBA/TPA-induced tumorigenesis using the constitutive K14-Cre Itga3KO model (Sachs et al., 2012) versus the inducible K19-Cre Itga3KO model (current study). When comparing the two models, the authors should also consider that in addition to directing $\alpha 3$ deletion to distinct compartments, $\alpha 3$ deletion is constitutive in the K14-Cre model (i.e., it occurs early in skin development) but induced in adult skin of K19-Cre mice just prior to tumor initiation. Thus, direct comparisons may be confounded by different timing of $\alpha 3\beta 1$ ablation that alters the microenvironment differently in the two models, or leads to compensatory changes in one model but not the other.

2. RNAseq (Fig. 6) revealed differential expression of only 15 genes in GFP+ Itga3KO and WT keratinocytes (one of which was Itga3 itself), a few of which encode secreted proteins. Only one secreted factor (CCN2) is validated, and it is neither directly linked to a change in the tumor microenvironment in vivo, nor shown to be required for skin tumorigenesis. Therefore, conclusions that $\alpha 3\beta 1$ broadly regulates the secretome or modulates the microenvironment (as stated in the title) should be tempered given the narrow scope of this study.

3. For different experiments, CRE induction is initiated in different aged mice, presumably at different hair cycle stages. Have the authors confirmed that $\alpha 3$ KO is comparable and similarly maintained (for long-term experiments) in HFs of K19 Itga3KO mice at different ages? What is the advantage of using a tamoxifen-inducible KO model rather than constitutive K19-CRE, given that results are compared to those from the previous study that used constitutive K14-CRE?

4. The hypothesis regarding a role for CCN2 is a bit unclear. Is CCN2 thought to be involved only in initial tumor formation, or also in tumor growth? It would be easy to assess CCN2 expression in skin tumors, as well as in tumors that do form in K19 $\alpha 3$ KO or K14 $\alpha 3$ KO mice. In lines 213-215, the authors conclude that "during the initiation of tumorigenesis $\alpha 3\beta 1$ in HB SCs suppresses HF differentiation and affects tumor environment through modulation of paracrine signaling, reflected in the expression of CCN2." Without experiments to directly test a role for CCN2 function in vivo, this statement should be tempered.

5. Fig. 7: The authors' hypothesis should also be bolstered by experiments that definitively link $\alpha 3\beta 1$ -dependent CCN2 secretion to 3D spheroid growth. For example, does treatment of either $\alpha 3$ KO cells or CCN2-deficient cells with exogenous CCN2, or with conditioned medium from $\alpha 3$ + keratinocytes, restore spheroid growth? Assays were not performed to rule out cell autonomous proliferation defects in CCN2 KO cells that may lead to smaller spheroids.

Minor comments:

6. In lines 127-129, the authors state that "no differences were observed in the length of neoepidermis three days after the wounding and in the percentage of closed wounds five days after the wounds were inflicted" between K19 Itga3KO and WT mice. Then in line 133-134 they describe a "small, but significant increase in the number of GFP-positive cells in the neoepidermis of K19 Itga3KO mice at the final stages of wound closure", followed by the statement that "this is consistent with previous observations that the absence of $\alpha 3\beta 1$ promotes cell migration during wound-healing, resulting in faster re-epithelization (Margadant et al., 2009)." These statements

appear contradictory. With regard to the second statement, the contribution of cells to neo-epidermis could differ in K19 Itga3KO skin without altering re-epithelialization rate, so it is not necessary to link these observations.

7. Fig. 1 data "demonstrate that $\alpha3\beta1$ plays no major role in HBs of adult mice under normal conditions", which appears in contrast with another group's report that $\alpha3\beta1$ is essential for hair follicle maintenance and morphogenesis (Conti et al., 2003, J. Cell Sci. 116). These distinct phenotypes could be due to differences in the genetic models, but the authors should mention this discrepancy.

8. Whole mounts of tail epidermis are used as a surrogate for back epidermis (i.e., wound healing and tumor studies are performed on the back), so it should be explained (or cited) how tail skin is a suitable surrogate for back skin.

9. Fig. 2d: FACS quantification of back skin epidermis includes cells of the IFE, so normalizing SC to total $\alpha3$ -positive cells is complicated by the fact that IFE cells in K14 Itga3KO mice do not contain $\alpha3$. It should be clarified how this normalization allows for assessment of absolute SC number. Otherwise, the reported "relative" number seems meaningless.

10. In Fig. 4a, differences in individual tumor size cannot be determined from this presentation. Is the difference in tumor burden seen in K19 Itga3 KO mice (right graph, total tumor volume/mouse) due simply to reduced tumor number (left graph)? Or are tumors in KO animals both fewer and smaller?

11. Lines 155-162: Regarding the increased aggressiveness of $\alpha3$ KO tumors compared with WT tumors (Fig. 4c-e), the authors explain that "yield...was too low for any firm conclusions", and there is no further expansion on this point. Since these data are not robust due to low sample number, and their interpretation relies heavily on previous publication in a different model, they would be more appropriate as supplemental data.

12. Regarding lines 164-166, is it accurate to state that total epidermal deletion of $\alpha3$ "completely" prevents tumor formation? (Line 160 refers back to tumors formed in K14 Itga3 KO mice.)

13. In Fig. 2b, it is difficult to discern the K15-positive cells to which arrowheads point. Enlargement may help, as would repeating arrowheads in the $\alpha3$ stained image to identify them as $\alpha3$ negative.

14. In Fig. 3b, the WT column is missing an error bar.

15. There are many mis-spellings/typos throughout the text.

16. The sentence that ends on line 238 needs a figure reference.

17. The sentence that ends on line 63 needs a citation.

18. Lines 753, 758: References should be to Supplemental Figs. 4a and 4c, respectively (not 5a and 5c).

Point-by-point response to the reviewers' comments

Reviewer 1

Minor points:

1. We agree with the reviewer that despite the fact that no experiments with CCN2 KO mice have been included in this study, the reported results are interesting and likely will spark researchers' interest to stimulate further research along this line.
2. The reviewer points out that SKH1 hairless mice still possess hair bulge stem cells (HB SCs) despite the lack of the hair growth in adult animals. In light of the studies by Miller et al. (Miller et al. 2001), Panteleyev et al. (Panteleyev et al. 1998) and Singh et al. (Singh et al. 2013) we agree that cells, positive for HB markers may be present in hairless mice and have amended our text accordingly: we have removed the sentence stating that the presence of tumorigenesis in hairless mice indicates the non-HB origin of tumors.
3. We agree that the long-term role of HB-originating cells in wound healing is somewhat controversial and have added the study by Sunny Wong to our manuscript (lines 44, 121). The differences in the contribution of hair bulge cells to neoepidermis formation and in their fate across different studies could to some extent originate from differences in experimental design, as pointed out by Garcin and Ansell (Garcin and Ansell 2017). As our study only investigates the efflux of the HB keratinocytes, which happens during the beginning of wound healing, rather than their long-term fate, we have amended the manuscript so that we avoid the speculations about the latter (lines 44, 121).

Reviewer 2

Regarding the general comments, we have edited the article for the grammatical and spelling mistakes and have strengthened the notion that $\alpha3\beta1$ -mediated secretion of CCN2 can affect tumorigenesis by demonstrating that the treatment with exogenous CCN2 promotes colony formation and spheroid growth of CCN2 KO transformed keratinocytes (Figure 8c-d and g-h, Supplementary fig S6b-d) (discussed in detail in the paragraph addressed to the Reviewer 3, point 5).

Specific comments:

1. Short-term DMBA/TPA treatment, such as DMBA-application followed by the 2 weeks of TPA-treatment that we have used in our study, or by 5 weeks of TPA-treatment as reported in the study, pointed out by reviewer, have been shown to suffice for the initiation of papilloma growth (Diwan et al. 1985; Hennings et al. 1985). Therefore, the epidermal changes at this point of two-stage chemical carcinogenesis reflect the cell environment that is able to support the outgrowth of papillomas. Our decision for short-term DMBA/TPA-treatment has been further supported by the finding that short-term TPA-treatment promotes the loss of slow-cycling label retaining cells in mice, lacking epidermal $\alpha3\beta1$ (Sachs et al. 2012), by our observation of activation of several signaling pathways that are known to be essential for the initiation papillomas at this time point (data in submission) and by the results of two-stage carcinogenesis treatment, which showed that deletion of *Itga3* in HB SCs primarily affects the initiation of tumorigenesis rather than later stages of tumor growth (Fig 4a and Supplementary fig S3a). As suggested, we have clarified our rationale behind the usage of the short-term DMBA/TPA treatment in Results section (lines 113-116 and 191-195). However, as the reviewer points out, it is important to note that stopping with the treatment after short-term DMBA/TPA application results in the reduced

number of papillomas and higher percentage of “high-risk” tumors (i.e. papillomas that will progress to become squamous cell carcinomas), as compared to long-term treatment (Darwiche et al. 2007; Diwan et al. 1985; Hennings et al. 1985). These “high-risk” papillomas have a distinct expression profile and show reduced T-cell infiltration (Darwiche et al. 2007). Compared to these studies, the crucial difference of the short-term DMBA/TPA treatment that we have applied is the final time point: whereas we investigated the skin immediately after the last TPA-treatment, previously mentioned studies examined skin several weeks after final TPA application, when “high-risk” papillomas that do not need continuous TPA-treatment for the outgrowth would form. As our short-term DMBA/TPA-treatment thus represents the continuous TPA-treatment, we would expect that the selection for the outgrowth of TPA-independent “high-risk” papillomas has not occurred yet and that the hyperplastic epidermis is at this point primed for the formation of “high” and “low” risk papillomas.

2. The reviewer pointed out an interesting study on the origin of the chemically-induced skin tumors that has escaped our attention; we have added it to our discussion section (lines 300-301, 306).

Reviewer 3

Specific comments:

1. Reviewer rightfully points out the limitation of our study, namely that we compare mouse models with inducible (K19-CreER) and constitutive (K14-Cre) promoters. We choose the inducible HB-targeted promoter to perform lineage tracing experiments, and it is agreeable therefore that we cannot formally exclude the possibility that inducing *Itga3* deletion at different mouse ages and development stages has resulted in cell-environmental or other differences between the two tumor models. However, the subtlety of the phenotype of constitutive *Itga3* epidermal deletion at the age when HB-specific *Itga3* deletion was induced in K19-driven mouse model offers some reassurance (Margadant et al. 2009). We have added this concern to the Discussion (lines 372-378).

2. In addition to the results of the RNAseq experiment, which we agree surprisingly has provided a low number of hits, our rationale behind the role of $\alpha3\beta1$ in modulation of microenvironment was also based on our observation that HB SCs hardly contribute to the tumor mass despite affecting tumor formation. Therefore, we reasoned that the most plausible explanation of how $\alpha3\beta1$ can indirectly affect tumorigenesis would be via modulation of microenvironment. However, it is indeed possible that HB-residing $\alpha3\beta1$ can affect tumor-initiating cells also via other means, such as affecting neighboring keratinocytes directly. Thus, we agree with the reviewer that the descriptions of the proposed role of $\alpha3\beta1$ in modulation of microenvironment and in regulation of HB secretome should be tempered. We have done this by changing the title and applying changes in following sections: Abstract (lines 33-35), Results (lines 186-188, 220-223) and Discussion (lines 321-322, 380-382).

3. Indeed, we induced the Cre-recombinase at 2 different mouse ages: for lineage tracing, we injected mice at P21 of the first telogen phase of the hair cycle, so that we could follow the HB-originating cells during the first hair cycle (until the second telogen phase at 7 weeks, i.e. P49), when hair cycle is still synchronized in the back skin of mice. As the experiments at this age were meant to be descriptive, we have not quantified the efficiency of *Itga3* deletion, but have used IF staining for $\alpha3$ in back skin and whole tail mounts to ensure that traced GFP-positive cells have been depleted of $\alpha3\beta1$ (Figure 1 c-e, Supplementary figure S1b). For wound healing and DMBA/TPA-treatment experiments (long and short-

term) we followed previous protocols, and thus started with experiments when mice were 7-week-old (P49, i.e. second telogen). To ensure efficient *Itga3* deletion, we have treated mice with tamoxifen approximately 1 week in advance (treatment for 4 days P40-P43). The deletion of $\alpha3\beta1$ following this treatment is assessed in 7-week-old mice in figures 1b and Supplementary figure S1a. The age of treated mice is described in the Materials and Methods section. We have clarified the treatment in figure legends. As mentioned under point 1, the primary reason for using the K19 promoter with inducible Cre was to perform lineage tracing experiments to assess a possible egress of $\alpha3$ -deficient HB SCs from hair follicles. As HB stem cells over time contribute to most populations of hair follicles, we reasoned that the lineage tracing of stem cells would only be possible with timed induction. Similarly, to visualize the role of HB SCs rather than other follicular cells during the long- and short-term tumorigenesis experiments, timed induction was deemed necessary.

4. The reviewer raises a justifiable observation that our first study did not have a clear hypothesis on the role of CCN2 in two-stage carcinogenesis. Therefore, we investigated the expression of CCN2 in available full-grown tumors, formed by two-stage tumorigenesis treatment. The finding that the expression of CCN2 in K19 *Itga3* WT papillomas and MSCC spheroids, grown for prolonged periods of time, is very low (Figure 8e, Supplementary Fig S4c), suggest that this protein likely has no role in the late stage of tumor growth. This hypothesis was further strengthened by our observation that the GFP-positive areas in tumors lacked CCN2 expression and thus that the CCN2-positive cells in full grown tumors are likely not associated with HB SCs (supplementary Fig S4d). The possibility that CCN2 has a pro-tumorigenic role at the initiation stage of tumorigenesis and/or during early tumor outgrowth was further suggested by *in vitro* data, which showed that the treatment with exogenous CCN2 had no effect on the growth of spheroids that already had been formed, whereas seeding cells in the presence of CCN2 promoted spheroid growth, albeit very moderately (Supplementary fig S6d). Based on these findings we have clarified our hypothesis and suggest that if the *in vitro* pro-tumorigenic role of CCN2 could be translated *in vivo*, it would likely take place during early tumorigenesis (lines 213-219, 342-346). Furthermore, in the absence of *in vivo* evidence for a pro-tumorigenic role of CCN2, we have tempered the concluding sentence in lines 220-223 as follows: "Together, this data shows that during the initiation stage of tumorigenesis $\alpha3\beta1$ in HB SCs suppresses HF differentiation and regulates the expression of CCN2 and several other proteins that are part of HB SC secretome. Thus, $\alpha3\beta1$ might affect the cell environment during early tumorigenesis through regulation of the paracrine signaling."

We should note that we were not able to assess the expression of CCN2 in K19 *Itga3* KO mice, as the samples were not yet ready when the closure of the research laboratories has been issued due to the Covid-19 pandemic. However, we think that the low CCN2 expression in the tumors of K19 *Itga3* WT mice, the lack of correlation between CCN2 expression and HB SCs and *in vitro* data mentioned above together provide support for our hypothesis that CCN2 does not play a role in late stages of tumor growth and as such we believe that determining the expression of CCN2 in the tumors, isolated from K19 *Itga3* KO mice would not provide additional crucial data. If such data is still deemed necessary, we can include it to our manuscript when restrictions due to the pandemic are lifted. As we do not yet know when this will happen, we are afraid that in such event, we would need to ask for more time to finish the revised manuscript.

5. As suggested, we have further bolstered the hypothesis that secreted CCN2 can promote pro-tumorigenic environment by treating CCN KO and *Itga3* KO MSCCs with exogenous CCN2. As can be seen in Figure 8c and g and Supplementary figure S6c, exogenous CCN2 increased colony formation of CCN2 KO clones and promoted their spheroid growth in 3D Matrigel, albeit the latter only to small extent, which suggests that there is also an autonomous effect of CCN2 deletion on spheroid growth. Interestingly, exogenous CCN2 could not promote the colony formation or 3D growth of *Itga3* KO keratinocytes (Figure 8d, h), which shows that CCN2 may enhance the tumorigenic potential of transformed keratinocytes but is not an essential component of the tumor promoting function of $\alpha3\beta1$.

Minor comments:

6. We agree with the reviewer that the link with the previous literature on the role of total epidermal deletion of *Itga3* in wound re-epithelization appears contradictory. Whereas epidermal *Itga3* deletion affected the rate of wound closure (Margadant et al. 2009), HB-specific deletion of *Itga3* in our study only affected the contribution of HB cells to neoepidermis but did not result in faster wound healing. However, as the reviewer already pointed out, it is hardly surprising that the differences in the contribution of HB SCs towards the neoepidermis did not affect the final rate of wound closure, considering that they constitute only a minor fraction of the newly formed epidermis. As suggested, we have avoided this apparent contradiction by removing the link between wound closure and $\alpha3\beta1$ -mediated cell migration during wound healing (lines 131-133).

7. As suggested, discrepancies in the literature on the role of *Itga3* deletion on the hair follicle homeostasis and growth are likely due to differences in mouse strains. Whereas our findings that the hair growth of mice, bred on FVB background (and of the age, used in our study), does not depend on $\alpha3\beta1$ is consistent with previous literature (Sachs et al. 2012), experiments with the mice bred on C57Bl/6 background showed otherwise (Conti et al. 2003; Sachs et al. 2012). We have added this to discussion (lines 294-297).

8. The miss-localized keratin15-positive keratinocytes have been previously most evidently demonstrated in the whole mounts of tail epidermis (Sachs et al. 2012), therefore, we used similar model for the conformation of this published data (Fig 2b). However, we agree that tail epidermis might not be completely comparable to the epidermis of the mouse back. Therefore, all our new experiments that have been carried out using tail whole mounts were repeated with sections of the back epidermis. We have clarified this in the text and figure legends. Lineage tracing of the HB-originating keratinocytes is presented in the whole mounts in figures 1c and supplementary figure S1b and in back skin in figure 1e. The localization of $\alpha3\beta1$ is investigated in the whole tail mounts in figure 2b and in back epidermis in figures 2c and 2d. Similarly, the quantification of the keratin 15 expression described in figure 2e was carried out in tail as well as back epidermis.

9. The goal of the experiment, presented in figure 2d, was to investigate whether the proportion of $\alpha3$ -positive keratinocytes that localize to hair bulges (i.e. CD34-positive cells) in the epidermis of KO mice is comparable to WT mice (indicating that the remaining $\alpha3$ -positive keratinocytes in K14 *Itga3* KO mice are randomly distributed over interfollicular epidermis and hair follicles), or not (indicating that remaining $\alpha3$ -positive keratinocytes in K14 *Itga3* KO mice distribute non-stochastically). Therefore, we

determined the proportion of CD34 and α 3-positive cells out of total α 3-positive population in several animals of both strains. As can be seen from the figure 2d and scatter plot in supplementary figure 2a, not all the remaining α 3-positive cells localize to HBs (based on CD34 marker), however, in K14 *Itga3* KO mice these HB-localized α 3-positive keratinocytes represent significantly bigger proportion of total α 3-positive cells compared to the WT mice. Therefore, the normalization on total α 3 staining was not carried out in order to assess absolute HB size, but to determine the percentage of α 3-positive cells that localize to HBs. We have clarified this in the text (lines 99-102) and in figure legend.

10. The reviewer raises a good point that from our representation of the data it was not possible to assess the differences in the individual tumor sizes. Therefore, the data was reanalyzed for the average tumor size per mouse and added to Supplementary figure S3a. As it can be seen from this figure, the deletion of *Itga3* from HBs mostly affects tumor outgrowth, as it results only in small, but significant reduction of the average tumor size.

11. As suggested, we have moved the data regarding the increased aggressiveness of K19 *Itga3* KO tumors to supplemental data (Supplementary figure S3b and S3c).

12. Indeed, several of K14 *Itga3* KO mice still developed sparse papillomas of no more than 3mm in diameter. Therefore, we have tempered the sentence in question to reflect this point (line 161).

13. Figure 2b has been modified as suggested.

14. Indeed, the WT bar in figure 3b does not have an error bar as all 4 mice had 25% of wounds closed at this time point. To clarify this, we have added the source data to the final submission of the paper.

15. The manuscript has been edited for spelling and grammatical mistakes.

16. The figure reference has been added (line 241).

17. The citation was added to the sentence (line 61).

18. The figure legend has been amended according to the revised figure.

ADDITIONAL COMMENTS

We have amended the graph in Figure 8b so that it represents the average values of technical replicates, as was already stated in the figure legends and which were already a source for statistical analysis. Wrong graph was imported in figure before, which represented data that equalized technical and biological replicates and thus did not reflect the figure legend or statistics used. Our apologies.

REFERENCES

Conti FJA, Rudling RJ, Robson A, Hodivala-Dilke KM. α 3 β 1-integrin regulates hair follicle but not interfollicular morphogenesis in adult epidermis. *J. Cell. Sci.* 2003;116(Pt 13):2737–47

Darwiche N, Ryscavage A, Perez-Lorenzo R, Wright L, Bae D-S, Hennings H, et al. Expression profile of skin papillomas with high cancer risk displays a unique genetic signature that clusters with squamous cell carcinomas and predicts risk for malignant conversion. *Oncogene*. 2007;26(48):6885–95

Diwan BA, Ward JM, Henneman J, Wenk ML. Effects of short-term exposure to the tumor promoter, 12-O-tetradecanoylphorbol-13-acetate on skin carcinogenesis in SENCAR mice. *Cancer Lett*. 1985;26(2):177–84

Garcin CL, Ansell DM. The battle of the bulge: re-evaluating hair follicle stem cells in wound repair. *Experimental Dermatology*. 2017;26(2):101–4

Hennings H, Shores R, Mitchell P, Spangler EF, Yuspa SH. Induction of papillomas with a high probability of conversion to malignancy. *Carcinogenesis*. 1985;6(11):1607–10

Margadant C, Raymond K, Kreft M, Sachs N, Janssen H, Sonnenberg A. Integrin alpha3beta1 inhibits directional migration and wound re-epithelialization in the skin. *J. Cell. Sci*. 2009;122(Pt 2):278–88

Miller J, Djabali K, Chen T, Liu Y, Ioffreda M, Lyle S, et al. Atrichia caused by mutations in the vitamin D receptor gene is a phenocopy of generalized atrichia caused by mutations in the hairless gene. *J. Invest. Dermatol*. 2001;117(3):612–7

Panteleyev AA, van der Veen C, Rosenbach T, Müller-Röver S, Sokolov VE, Paus R. Towards Defining the Pathogenesis of the Hairless Phenotype. *Journal of Investigative Dermatology*. 1998;110(6):902–7

Sachs N, Secades P, van Hulst L, Kreft M, Song J-Y, Sonnenberg A. Loss of integrin $\alpha 3$ prevents skin tumor formation by promoting epidermal turnover and depletion of slow-cycling cells. *Proc Natl Acad Sci U S A*. 2012;109(52):21468–73

Singh A, Singh A, Verma AK. Abstract 5020: Characterization of epidermal stem cells in SKH1 hairless mice, a widely used mouse model to investigate ultraviolet radiation carcinogenesis. *Cancer Res*. 2013;73(8 Supplement):5020–5020

April 30, 2020

RE: Life Science Alliance Manuscript #LSA-2020-00645RR

Prof. Arnoud Sonnenberg
Netherlands Cancer Institute
Division of Cell Biology
Plesmanlaan 121
Amsterdam 1066 CX
Netherlands

Dear Dr. Sonnenberg,

Thank you for submitting your revised manuscript entitled "Integrin $\alpha 3\beta 1$ in hair bulge stem cells modulates CCN2 expression and promotes skin tumorigenesis". As you will see, the reviewers appreciate the changes introduced in revision, and we would thus be happy to publish your paper in Life Science Alliance pending final minor revision:

- please address the remaining reviewer comments
- please add a scale bar to figure 2c
- please check your statistical analysis again (e.g., fig 7 and 8, which should probably use different tests)
- please include a statement on the approval of the mouse work in your M&M section (please see our guidelines)
- It would be a shame not to suggest a cover image in your case - we would need a high resolution, raw image for this. You can upload them directly into our submission system or request a link from me in case the files are too big. I think particularly images similar to those in Fig 1c, 1d or S1b would lend themselves nicely.

A. FINAL FILES:

B. MANUSCRIPT ORGANIZATION AND FORMATTING:

Thank you for your attention to these final processing requirements.

Sincerely,

Andrea Leibfried, PhD
Executive Editor
Life Science Alliance
Meyerhofstr. 1
69117 Heidelberg, Germany
t +49 6221 8891 502

Reviewer #1 (Comments to the Authors (Required)):

The authors adequately addressed all of my concerns in the revised manuscript.

Reviewer #2 (Comments to the Authors (Required)):

These authors found that hair bulge stem cells are likely not the cancer cells of origin in DMBA/TPA cutaneous carcinogenesis; however, HB cells contribute to the microenvironment through $\alpha3\beta1$ modulated CCN2 expression. In this revised manuscript the authors have satisfactorily addressed my queries and comments from the previous review. This paper is a significant and interesting advance to the field, in part because no one has yet offered an explanation as to why Li et al. found so few HB progeny in the papillomas, and that the actual role of the HB cells may be the contribution of a tumor-conducive microenvironment. The authors have also satisfactorily addressed the comments of the other reviewers. The authors are commended for their beautiful and informative photomicrography.

Reviewer #3 (Comments to the Authors (Required)):

In this revised manuscript the authors have addressed all previous concerns through clarification in the writing or inclusion of new data. These changes have resulted in a more clearly presented study that will be an important contribution to the field with regard to cellular origins of skin tumors and the role that integrin $\alpha3\beta1$ plays in this process. Below are a few corrections/suggestions for the authors to consider.

1. Line 156: The statement that 3 out of 7 WT tumors progressed is discrepant with the associated pie chart (Fig. S3c) that indicates 4 progressed tumors.
2. Line 254: Would the word "sufficient" be more appropriate than "required"?
3. Line 319: Delete the word "in".
4. Line 332: Correct capitalization of the name "NEAGU" here, and of all authors in the Reference list.
5. Fig. S4 legend, line 888: Would the word "area" be better than "surface"?
6. Fig. S6 legend, line 901: The reference to Fig 8 is missing (i.e., figure 8c and 8d).

Dear Editor,

We are glad to hear that our revisions have been well received and that the manuscript is almost ready for the publication. As requested, we have addressed the final minor revisions and provide point-by-point response as follows:

Editor:

1. We have added the missing scale bars to the Figure 2c.
2. We have re-analyzed the experimental datasets where we compare more than two groups (i.e. Figure 7a-b, 8b-d, 8f-h, Supplementary figure S5a, Supplementary figure S6c-d). One-way ANOVA was used to analyze experiments with more than two groups and two-way ANOVA was performed on experimental data where we investigated the effect of two factors: different treatments and two distinct cell types. Type I errors were reduced by testing only planned comparisons among a relatively small number of means. Planned comparisons were conducted using Fisher's Least Significant Difference test after a global ANOVA was determined to be significant. We have accordingly adjusted the "Statistics" section in the Materials and Methods as well as the corresponding figure legends. As the usage of statistical analysis on small number of samples can be controversial, we have changed the plots in figures 7 and 8 where $n \leq 5$ to show raw data points with mean and standard deviation, so that the readers can draw their own conclusions independently of the statistical tests applied (Figures 7a, 8b-d).
3. We have added the required declaration to the Materials & Methods section (Generation of mice).
4. We have included a suggestion for the cover image.

Reviewer 1 and 2:

no additional comments to address

Reviewer 3:

1: The chart in Supplementary figure S3c represents the number and type of progressed tumors rather than mice. Five K19 *Itga3* KO mice and two K19 *Itga3* WT mice developed only one progressed tumor upon extended treatment, whereas one K19 *Itga3* WT mouse developed two progressed tumors. We realize that this was not clear and have thus added additional explanation to the figure legend of the Supplementary figure S3c.

2: We agree that "sufficient" would be more appropriate and have adjusted the sentence accordingly

3: "in" has been deleted

4: Capitalization has been corrected

5: As suggested, the word "surface" has been replaced with "area"

6: The missing "8" has been added

Thank you for assistance and handling our manuscript.

With kind regards,

Arnoud Sonnenberg

May 6, 2020

RE: Life Science Alliance Manuscript #LSA-2020-00645RRR

Prof. Arnoud Sonnenberg
Netherlands Cancer Institute
Division of Cell Biology
Plesmanlaan 121
Amsterdam 1066 CX
Netherlands

Dear Dr. Sonnenberg,

Thank you for submitting your Research Article entitled "Integrin $\alpha3\beta1$ in hair bulge stem cells modulates CCN2 expression and promotes skin tumorigenesis". It is a pleasure to let you know that your manuscript is now accepted for publication in Life Science Alliance. Congratulations on this interesting work.

DISTRIBUTION OF MATERIALS:

Again, congratulations on a very nice paper. I hope you found the review process to be constructive and are pleased with how the manuscript was handled editorially. We look forward to future exciting submissions from your lab.

Sincerely,
